# In a Nutshell, the Human Asked for This: Latent Goals for Following Temporal Specifications

**Borja G. León[1] & Murray Shanahan[1] & Francesco Belardinelli [1,2]**
[1] Department of Computing, Imperial College London, London, United Kingdom
[2] IBISC, Université d'Evry, Evry, France
`{borjagleon, m.shanahan, francesco.belardinelli}` *@imperial.ac.uk*

## Abstract

We address the problem of building agents whose goal is to learn to execute out-of distribution (OOD) multi-task instructions expressed in temporal logic (TL) by using deep reinforcement learning (DRL). Recent works provided evidence that the agent's neural architecture is a key feature when DRL agents are learning to solve OOD tasks in TL. Yet, the studies on this topic are still in their infancy. In this work, we propose a new deep learning configuration with inductive biases that lead agents to generate latent representations of their current goal, yielding a stronger generalization performance. We use these latent-goal networks within a neuro-symbolic framework that executes multi-task formally-defined instructions and contrast the performance of the proposed neural networks against employing different state-of-the-art (SOTA) architectures when generalizing to unseen instructions in OOD environments.

## 1 Introduction

Building agents that generalize their learning to satisfy new specifications is a crucial goal of artificial intelligence (Oh et al., 2017; Bahdanau et al., 2019; Hill et al., 2021). Deep reinforcement learning (DRL) holds promise in autonomous agents that tackle complex scenarios (Silver et al., 2017; Samvelyan et al., 2019), which motivates ongoing research with DRL algorithms following human instructions expressed in natural language (Yu et al., 2018; Mao et al., 2019). Unfortunately, generalization in DRL is linked to training autonomous agents with large numbers of instructions requiring to build manually those natural language expressions and their corresponding reward functions, which prevents such methods from scaling efficiently (Lake, 2019; Vaezipoor et al., 2021).

These considerations inspired research on agents learning from formally specified instructions (Wen & Topcu, 2016; Alshiekh et al., 2018; Simão et al., 2021) as a substitute for natural language. Formal languages (Huth & Ryan, 2004) offer desirable properties such as unambiguous semantics and compositional syntax, allowing to automatically generate large amounts of training instructions and their corresponding reward functions. Earlier contributions in this area rely on the compositional nature of formal languages, often employing multiple policy networks to execute temporal logic (TL) instructions (Andreas et al., 2017; Icarte et al., 2018; Kuo et al., 2020). However, these approaches do not scale well with the number of instructions since policy networks are a computationally costly resource and, consequently, these earlier studies are restricted to relatively small state-spaces environments that require less computation, e.g., non-visual settings. More recent contributions (León et al., 2020; Vaezipoor et al., 2021) have presented DRL frameworks that are capable of generalizing to large numbers of out-of-distribution (OOD, i.e., never seen during training) instructions while relying on a single policy network. Those latter studies evidence that deep learning architectures within DRL agents are key towards the ability of agents to generalize formal instructions.

However, network architectures in previous literature have mostly followed a standard configuration, where all the network's layers except for those encoding the input (e.g., convolutional layers for images, recurrent layers for text) have equal access to both human instructions and environment's observation. We propose a novel configuration to help agents generalize better by having a

task-agnostic representation of the current state concatenated to a latent goal that is form by processing both observation of the agent and human instruction. As a motivation example, let us consider two scenarios: 1) a robot that is at the center of an empty room with a red square at the bottom right corner, 2) a robot at the same position in a room that is identical to the one in (1) except that the red square is now green. Intuitively, we can say that, if we give the instruction "go to the red square" in (1), the goal of the robot is the same as if we say "go to the green square" in (2), because the two tasks can be abstracted as "turn to face the bottom right of the room, then move straight". In a nutshell, computing the human instruction together with the current state of the environment – being at the center of the room, with the object asked by the human in the bottom right corner – allows to deduce that the optimal policy is the same in both scenarios. Our **contributions** are listed as follows:

- We propose a new deep learning configuration (the latent-goal architecture) that helps agents to generalize better when solving multi-task instructions in OOD scenarios.
- We are first to evaluate the performance of multiple state-of-the-art (SOTA) neural networks targeting generalization when following temporal logic instructions.
- Through ablation studies, we find that networks generating independent pieces of information within their output are specially benefited from being used within latent-goal architectures.

The remaining of the paper is structured as follows: Section 2 briefly introduces the key concepts needed to follow this work. Section 3 presents the formal language used to generate instructions and the neuro-symbolic framework that agents use in our experiments. Then, Section 4 details the proposed network configuration. Section 5 includes the experimental settings, empirical results and ablation studies. Last, Sections 6 and 7 discuss related works and conclusions, respectively.

## 2  BACKGROUND

We develop agents aimed to execute OOD instructions expressed in TL while navigating in partially observable (p.o.) environments. Specifically, agents operate with a fixed perspective and a limited visual range. Below we introduce the basic concepts used in the rest of the paper.

**Reinforcement learning.**   Our p.o. environment is modelled as a p.o. Markov decision process (POMDP). A POMDP is a tuple $\mathcal{M} = \langle S, A, P, R, Z, O, \gamma \rangle$ where (i) $S$ is the set of *states* $s, s', \dots$; (ii) $A$ is the set of *actions* $a, a', \dots$; (iii) $P : S \times A \times S \to [0, 1]$ is the *(probabilistic) transition function*; (iv) $R : S \times A \times S \to \mathbb{R}$ is the *reward function*; (v) $Z$ is the set of *observations* $z, z', \dots$. (vi) $O : S \times A \times S \to Z$ is the *observation function*. (vii) $\gamma \in [0, 1)$ is the *discount factor*. At each time step $t$, the agent chooses an action $a_t \in A$ triggering an update in the current state from $s_t$ to $s_{t+1} \in S$ according to $P$. Then, $R_t = R(s_t, a, s_{t+1})$ provides the reward associated with the transition, and $O(s_t, a_t, s_{t+1})$ generates a new observation $o_{t+1}$ for the agent. Intuitively, the goal of the learning agent is to choose the policy $\pi$ that maximizes the expected sum of discounted rewards: $V^\pi(x) \stackrel{\text{def}}{=} \mathbb{E}_\pi \left[ \sum_{t \geq 0} \gamma^t r_t \right]$, where $\gamma \in [0, 1)$ is the discount factor (see Sutton & Barto (2018)).

**Relational networks.**  Relational networks (RelNets) are a particular kind of neural network whose structure is explicitly designed for reasoning about relations (Kemp & Tenenbaum, 2008). Previous contributions on solving formal instructions with DRL have been mainly focused on multi-layer perceptrons (MLPs) and recurrent networks (Goodfellow et al., 2016), yet RelNets are of particular interest in our context since they hold promise to improve generalization in DRL (Santoro et al., 2018). Their central principle is to constrain the functional form of the neural network in the relational module so that it captures the core common properties needed to reason about the relations between entities (e.g., objects) and their properties (e.g. color), see Santoro et al. (2017). Recently, relational networks have been enhanced with the inclusion of the widely-known key-value *attention* mechanism (Bahdanau et al., 2015). This method relies on the computation of attention scores that are used to decide in which portions of the input data the layer using this mechanism should focus on.

## 3  LEARNING TO SOLVE SATTL INSTRUCTIONS

In this section we detail the shared features of our agents. Concretely, Sec. 3.1 introduces the formal language we use to procedurally generate instructions, while Sec. 3.2 details the neuro-symbolic framework that we use while testing the different neural network configurations.

## 3.1 SAFETY-AWARE TASK TEMPORAL LOGIC

We investigate agents learning compositionally instructions in temporal logic. Below, we formally define the language from which we generate instructions to be executed by our agents. Specifically, we extend task temporal logic (TTL), defined in León et al. (2020) to study the ability of neural networks to learn systematically from instructions about reachability goals such as "eventually reach a sword". In this work, we evaluate agents that handle safety constrains as well, e.g. "walk on soil or stone until you reached a sword". Thus, we extend TTL to safety-aware task temporal logic (SATTL).

**Definition 1** (SATTL). *Let $AP$ be a set of propositional atoms. The sets of literals $l$, atomic tasks $\alpha$, and temporal formulas $T$ in SATTL are inductively defined as follows:*

$$
\begin{aligned}
l &\ ::=\ +p \mid -p \mid l \vee l \\
\alpha &\ ::=\ lUl \\
T &\ ::=\ \alpha \mid T;T \mid T \cup T
\end{aligned}
$$

Literals $l$ are positive ($+p$) or negative ($-p$) propositional atoms, or disjunctions thereof. Atomic tasks $\alpha$ are obtained by applying the temporal operator *until* ($U$) to pairs of literals. An atom $\alpha = lUl'$ is read as *it is the case that $l$ until $l'$ holds*. Temporal formulas $T$ are built from atomic tasks by using sequential composition (;) and non-deterministic choice ($\cup$). Note that $U$ and $\cup$ are different operators. We use an explicit positive operator ($+$) so that both positive and negative tasks have the same length. This is beneficial for learning negative literals ($-p$) when instructions are given visually, as highlighted in (León et al., 2020). The temporal operators *eventually* $\diamond$ and *always* $\square$ can be respectively defined as $\diamond l \equiv \text{true} U l$ and $\square l \equiv lUend$, where $end$ is a particular atom true only at the end of the episode. As TTL, SATTL is interpreted over finite traces $\lambda$, which in this context refers to finite sequences of states and actions. We denote time steps, i.e., instants, on the trace as $\lambda[j]$, for $0 \leq j < |\lambda|$, where $|\lambda|$ is the length of the trace. While, $\lambda[i,j]$ is the (sub)trace between instants $i$ and $j$. A *model* is a tuple $\mathcal{N} = \langle \mathcal{M}, L \rangle$, where $\mathcal{M}$ is a POMDP, and $L : S \to 2^{AP}$ is a labelling of states in $S$ with atoms in $AP$.

**Definition 2** (Satisfaction). *Let $\mathcal{N}$ be a model and $\lambda$ a finite path. We define the satisfaction relation $\models$ for literals $l$, atomic tasks $\alpha$, and temporal formulas $T$ on path $\lambda$ as follows:*

$$
\begin{aligned}
(\mathcal{N}, \lambda) &\models +p &&\text{iff} &&p \in L(\lambda[0]) \\
(\mathcal{N}, \lambda) &\models -p &&\text{iff} &&p \notin L(\lambda[0]) \\
(\mathcal{N}, \lambda) &\models l \vee l' &&\text{iff} &&(\mathcal{N}, \lambda) \models l \text{ or } (\mathcal{N}, \lambda) \models l. \\
(\mathcal{N}, \lambda) &\models lUl' &&\text{iff} &&\text{for some } 0 \leq j < |\lambda|, (\mathcal{N}, \lambda[j, |\lambda|]) \models l', \text{ and for every } t \in [0, j), \\
& && &&(\mathcal{N}, \lambda[t, j-1]) \models l \\
(\mathcal{N}, \lambda) &\models T;T' &&\text{iff} &&\text{for some } 0 \leq j < |\lambda|, (\mathcal{N}, \lambda[0, j]) \models T \text{ and } (\mathcal{N}, \lambda[j+1, |\lambda|]) \models T' \\
(\mathcal{N}, \lambda) &\models T \cup T' &&\text{iff} &&(\mathcal{N}, \lambda) \models T \text{ or } (\mathcal{N}, \lambda) \models T'
\end{aligned}
$$

Intuitively, by Def. 2 an atomic task $\alpha = c_\alpha U g_\alpha$ is satisfied if the "safety" condition $c_\alpha$ remains true until goal $g_\alpha$ is fulfilled. In the context of this work we are not interested in strict safety warranties but in agents trying to reach a goal while aiming to minimize the violation of additional conditions. Thus, we may say our agent has satisfied $\alpha$ even though it has not fulfilled $c_\alpha$ at every time step before $g_\alpha$. Formally, in those cases we consider traces that are not starting at the beginning of the episode, but from the state after the last violation. The agent is penalised for this behaviour through the reward function. Examples of tasks we use are $\alpha_1 = -\text{grass} U (+\text{axe} \vee +\text{sword})$ "avoid grass until you reach an axe or a sword" and $\alpha_2 = (+\text{soil} \vee +\text{mud})U\text{axe}$ "move through soil or mud until reaching an axe". It is not difficult to prove that SATTL is a fragment of the well-known Linear-time Temporal Logic over finite traces (LTL$_f$) (De Giacomo & Vardi, 2013). We provide a translation of SATTL into LTL$_f$ and the corresponding proof of truth preservation in Appendix F.

## 3.2 NEURO-SYMBOLIC AGENTS

We extend the neuro-symbolic framework from León et al. (2020), intended for DRL agents solving unseen TTL instructions, and adapted it to work with SATTL. In this section, we provide details about this framework, that entails a symbolic and a neural module. Intuitively, given a formal instruction $T$ in SATTL, the symbolic module (SM) decomposes $T$ into a sequence of atomic tasks $\alpha$ to be solved sequentially by the neural module (NM), which is instantiated as a DRL algorithm.

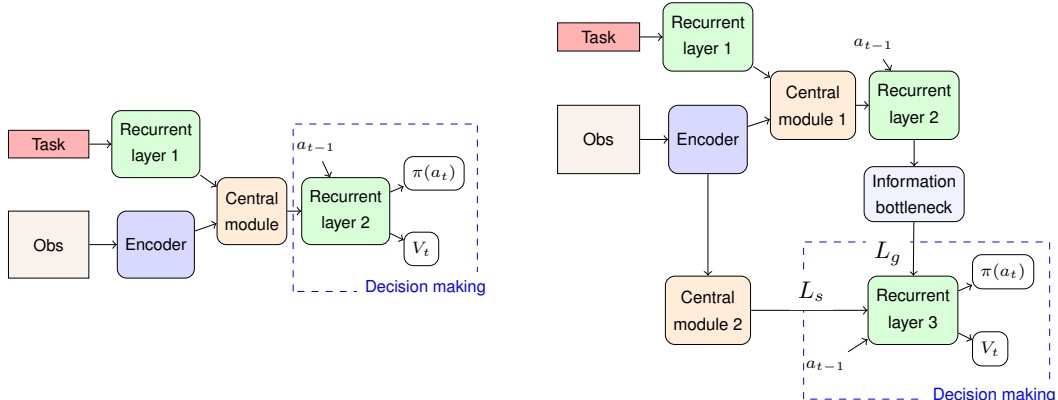

Figure 1: Neural network architectures. **Left:** standard architecture from previous literature. The central module can be either fully-connected or relational layers. Outputs $\pi(a_t)$ and $V_t$ refer to the actor and critic respectively (Mnih et al., 2016). **Right:** proposed latent-goal architecture.

**Symbolic module.**   Formally, the first component of the SM is the *extractor* $\mathcal{E}$, which transforms the complex formula $T$ into a list $\mathcal{K}$ consisting of all the sequences of atomic tasks $\alpha$ that satisfy $T$. As it is common in the literature (Kuo et al., 2020; Vaezipoor et al., 2021), the SM have access to an internal labelling function $\mathcal{L}_{\mathcal{I}} : Z \rightarrow 2^{AP}$, where $\mathcal{L}_{\mathcal{I}}$ is the restriction of $\mathcal{L}$ to the observation space and maps the observation into the set of atoms (e.g, reached_sword is an atom). Intuitively the labelling function acts as an *event detector* that fires when the propositions in $AP$ hold in the environment. The second functionality is the *progression function* $\mathcal{P}$, which given the output of $\mathcal{L}_{\mathcal{I}}$ and the current list $\mathcal{K}$, selects the next task $\alpha$ for the NM and updates $\mathcal{K}$. Once $\alpha$ is selected, the NM follows the given instruction until it is satisfied or until the time horizon is reached, while the role of the SM is to evaluate the fulfillment of the task based on the signals from $\mathcal{L}_{\mathcal{I}}$. From the outputs of $\mathcal{L}_{\mathcal{I}}$, the SM generates a reward function $R$, which provides the reward signal for the NM. Note that from the NM perspective, this signal is the standard reward signal of a POMDP – but in our case associated to the instruction–. Given a task $\alpha$, $R$ is defined so that the agent receives a positive reward ($+1$) if the goal condition becomes true ($\mathcal{L}_I(z_t) = g_\alpha$), a strong penalisation ($-1$) if the agent is neither satisfying the goal nor the safety condition ($\mathcal{L}_I(z_t) \neq g_\alpha$ or $c_\alpha$), and a small penalisation ($-0.05$) if none of the previous is true. The latter penalisation is meant to encourage the agent to reach the goal. The pseudocode of the SM and its functions $\mathcal{E}$ and $\mathcal{P}$ are included in Appendix A. Since the inner functioning of our symbolic module is analogous to the one in León et al. (2020), where authors show how the SM can process complex formulas $T$ given a NM that has learnt to execute tasks $\alpha$, we devote the remaining sections to the role of the deep learning architecture within the NM when solving atomic tasks, which are the main scope of this work (see end of Sec. 3.1 and Figure 2 for examples of atomic tasks).

**Neural module.**   The NM consists of a DRL algorithm interacting with the environment to maximize the discounted cumulative reward from $R$. All our agents use the same A2C algorithm, a synchronous version of A3C (Mnih et al., 2016). To facilitate a fair comparison, all the deep learning architectures use the same visual encoder and output layers for the respective benchmarks. Appendix C provides further detail about these features. As anticipated, the core of our approach is the deep learning architecture, detailed below.

## 4   LATENT-GOAL ARCHITECTURES

We presented the language employed to generate instructions (Sec 3.1) and the neuro-symbolic framework that use our agents (Sec 3.2). Now we move towards the main focus of this work, presenting a new neural network configuration that improves OOD generalization when executing novel multi-task instructions.

Literature about autonomous agents with generalization-specialized architectures such as relational deep layers (Shanahan et al., 2020) or modular hierarchical networks (Mittal et al., 2020), commonly

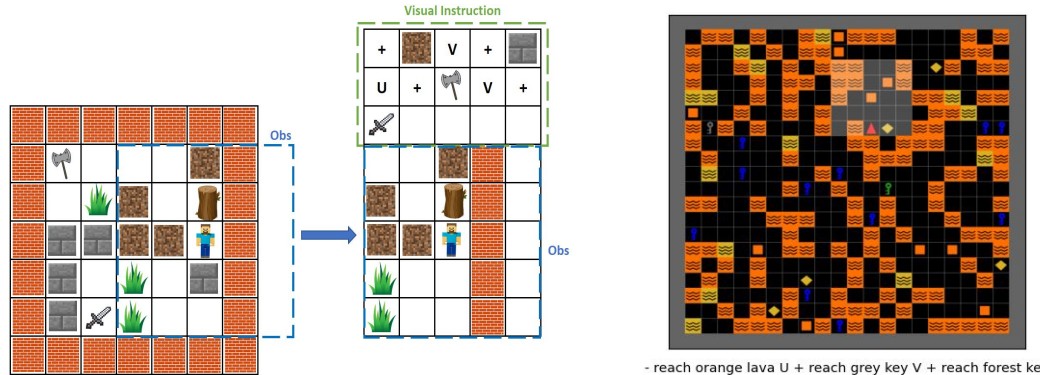

Figure 2: **Left:** a 7x7 Minecraft map (left) and its corresponding extended observation (right) with the visual instruction "move through either soil or stone until reaching an axe or a sword". Tasks are specified at the top of the extended observation by depicting objects and SATTL operators. Each tile has a resolution of 9x9 pixels. **Right:** a 22x22 map in MiniGrid where the agent needs to "avoid orange lava until reaching a gray or dark-green key". Instructions are given through a text channel and each tile has 8x8x3 pixels. Highlighted cells indicate the field of vision of the agent.

follows a standard model architecture set-up where the output of a visual encoder is passed through a central module (CM), which in our context is either a relational layer or an MLP, followed by a recurrent layer. Unfortunately, deep learning agents struggle to generalize instructions compositionally (Lake, 2019), even when generalizing between task-similar instructions.

Consequently, we propose an architecture configuration that induces agents to compute both human instructions and sensory inputs in a dedicated channel, generating a latent representation of the current goal. We call this type of deep learning model a *latent-goal architecture*. Figure 1 right shows our proposed configuration when instructions are given as text; while Figure 1 left illustrates the standard architecture from previous literature for comparison (Hill et al., 2020; Küttler et al., 2020; Vaezipoor et al., 2021). In our configuration, the key feature is that the decision-making layers receive two input streams $L_g$ and $L_s$. $L_g$ refers to the latent goal, i.e., a representation of the agent's current goal given the state and the human instructions, and $L_s$ is a vector that contains information about all sensory input processed by the agent except for the instruction. For a latent-goal architecture to generalize effectively there are two central design properties: i) $L_s$ should be task-agnostic, i.e., the layers computing $L_s$ should have no access to what the current task is. ii) $L_g$ should be passed through an information bottleneck, e.g., a low-dimensional fully-connected layer (FC). Condition i) forces the layers computing $L_s$ to generate representations of the states as generic as possible. By granting no access to the current task, these layers cannot discern which elements provide penalisation or rewards, encapsulating information in a fashion that facilitates generalization when executing novel instructions. Condition ii) ensures that decision-making layers rely on the generic representation from $L_s$ to extract most of the data from the environment, while the layers processing $L_g$ should focus on providing enough information about the goal.

## 5 EXPERIMENTS

Here we start presenting the experimental settings and continue with the main experimental results that highlighting the benefits of latent-goal architectures. We finish with ablation studies to better understand the inner working of the novel configuration.

**Empirical settings.** We use two benchmarks. One is a Minecraft-inspired environment – Minecraft for short – widely used in the RL-TL literature (Andreas et al., 2017; Vaezipoor et al., 2021). Particularly, we use the variant from León et al. (2020) that works with procedural maps and instructions. In this setting, instructions are visually given as part of the observation. The second benchmark is MiniGrid (Chevalier-Boisvert et al., 2018), which is also a procedural benchmark where instructions are given as text through a separated channel. Agents are trained in maps of random size $n \times n$, for $n = [7 - 10]$, and evaluated in maps of size $n = 7, 14, 22$. Figure 2 illustrates maps from

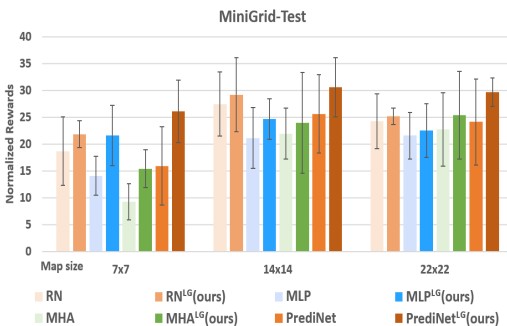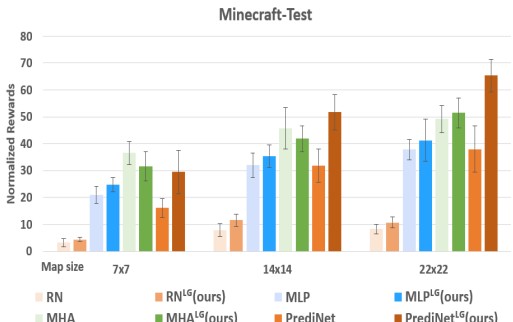

Figure 3: Results with OOD instructions in 500 maps (per size) of different dimensions. Sizes 14 and 22 are OOD. MC refers Minecraft whereas MG to MiniGrid. Results show the average reward and standard deviation from 10 independent runs (i.r.). Values are normalized so that 100 refers to the highest performance achieved by the best run globally in maps of the given size and benchmark.

both benchmarks. Instructions are SATTL tasks (see $\alpha_1$ or $\alpha_2$ in Sec. 3.1 for examples). We evaluate generalization to unseen instructions, thus results under label "Train", refer to performance with training instructions, while "Test" to OOD instructions. In MiniGrid, where agents require grounding words to objects, OOD specifications are formed with combinations of symbols and objects that were explicitly excluded from training, e.g., if an OOD instructions has the safety condition "avoid orange lava" it means that both color "orange" and shape "lava" where only present in training instructions of the type "reach orange lava". In Minecraft, where tasks are depicted with objects themselves, OOD instructions refer to tasks with zero-shot, i.e., never seen, objects. Since specifications are human-given, we assume access to the resolution of the visual instructions and select an encoder generating a visual embedding where instructions are independent from the rest the observation. This allows the separation of the task from the rest of the observation, which is beneficial for the latent-goal architectures that generate task-agnostic streams of data. The separation also facilitates a fair comparison between the text and visual instruction domains. Both benchmarks have at least 200K tasks for training and 65K OOD specifications. Further detail is given in Appendix B.

**Networks and baselines.** All the networks follow either a standard or a latent goal (LG) configuration (see Sec. 4). The encoder is a convolutional neural network (CNN) (LeCun et al., 2015) while the recurrent layer in the output is either a long short-term memory (LSTM) (Hochreiter & Schmidhuber, 1997), that we use in multi-layered architectures, or a bidirectional recurrent independent mechanism (BRIM) in the case of hierarchical models (Mittal et al., 2020). In MiniGrid, where instructions are given as text, we use a bidirectional LSTM (Schuster & Paliwal, 1997). We consider four variants for the central modules within the architectures (with detailed information given in Appendix C.3):

- *Relation net* (RN) from (Santoro et al., 2017), designed to dispense FC and LSTM layers with relational reasoning.
- *Multi-head attention net* (MHA) (Zambaldi et al., 2018), which combines relational connections with key-value attention.
- *PrediNet* (Shanahan et al., 2020), also combining relational and attention operations but specifically designed to form propositional representations of the output.
- *MLP* consisting of a single FC layer, which is meant to act as baseline against the relation-based modules.

**Results with multi-layer architectures.** Fig. 3 shows the performance of the different multi-layer networks (see Fig. 4 for the reference of a random walker). Results that are labelled as CM networks (e.g., "RN") refer to standard configurations, whose CM uses the network of the label. Correspondingly, results with the LG superscript (e.g., RN$^{LG}$) refer to latent-goal configurations. Overall, we see that the latent-goal architecture improves the performance of all the networks in most scenarios. And is always the best performer in the hardest setting ($n = 22$), where the best latent-goal model achieves 33% and 22% better performance compared to the best standard architectures in Minecraft and MiniGrid, respectively. Training results and tables with the specific values are given in

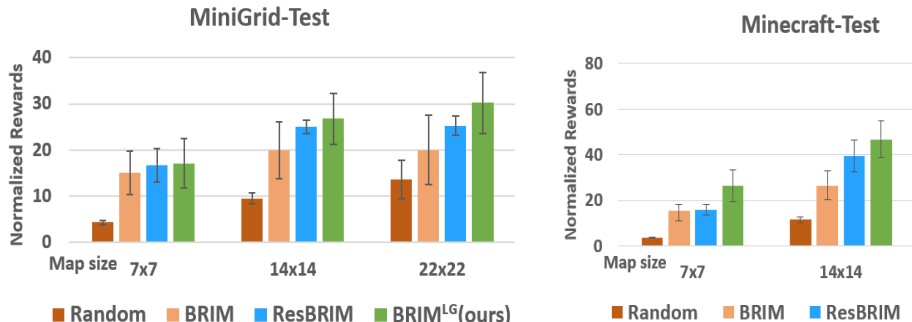

Figure 4: Results of 5 i.r. of vanilla BRIM (BRIM), BRIM with residual connections (ResBRIM), and with LGs (BRIM$^{LG}$).

Appendix D, together with a two-way analysis of variance (ANOVA, independent variables being network architecture in the central modules, and the architecture configuration) that also confirms the statistical impact of the latent-goals. We also observe that the PrediNet is the network that benefits the most from the new configuration (up to 74%) and PrediNet$^{LG}$ becomes clearly the best architecture. Noticeably, the two most different results with latent-goal architectures come from the networks that combine relational and attention mechanisms (MHA and PrediNet). This point if further examined in the ablation below.

**Results with hierarchical architectures.** We demonstrate that latent goals can improve the performance of hierarchical architectures. Specifically, in line with the two design conditions highlighted in Sec. 4, we evaluate empirically a hierarchical architecture that restricts the amount of information shared between layers of different hierarchy, while providing instruction and observation data to the bottom layer of the architecture and a task-agnostic input of the environment to the top-level of the hierarchy (see Fig. 8 in the Appendix for an illustration). In these experiments we fix the PrediNet as the default network for the different CMs – note that the PrediNet$^{LG}$ achieved the best results in the previous evaluation – and replaced the last recurrent layer of the architectures with a BRIM. BRIMs are modularized hierarchical networks that hold promise for OOD generalization in RL. A particularly interesting feature of BRIMs for our method is that they employ a sparse attention mechanism to communicate between different hierarchies of layers. Specifically, each layer is composed of various modules, and at each time step only the top $k$ modules (according to a key-value attention mechanism) will be able to communicate with modules from different layers, where $k$ is a hyperparameter. Thus, when using latent goals with BRIMs we remove the FC layer that acted as information-bottleneck above. An illustration of the architectures using BRIMs is given in Appendix C.3. We do not perform any hyperparameter optimization with BRIMs and use directly the same configuration from the RL experiments in Mittal et al. (2020). Fig. 4 contrasts the performance when using a vanilla BRIM layer (BRIM) against using a residual connection (He et al., 2016) (ResBRIM) or latent goals (BRIM$^{LG}$). We see that inducing the latent goals improves the performance of the hierarchical network in all settings. Hence, we see that deep hierarchical architectures can also capitalize on specialized hierarchy levels where bottom layers produce latent representations of the goal while the higher levels receive task-agnostic information of the state.

## 5.1 ABLATIONS

Contrasting MHA and PrediNet with their latent-goal counterparts, we see that the particular differences of the PrediNet with respect to the MHA internal functioning benefits the proposed latent-goal architecture. After a closer inspection, we see that the most significant difference comes from a channeling of the input within the PrediNet. This network encourages a kind of the semantic separation of the representations that it learns, making an element-wise subtraction of the extracted features, an operation not found in the MHA. Additionally, we see that the PrediNet modifies its output to explicitly represent positions of the pair of features selected by each of its heads, also not present in the MHA. Fig. 5 inspects the impact of individually removing these distinct features within the PrediNet and whether the PrediNet is beneficial in both CMs or it is better to use an MHA in any of the modules. We find that the element-wise subtraction is the key feature within the PrediNet since its

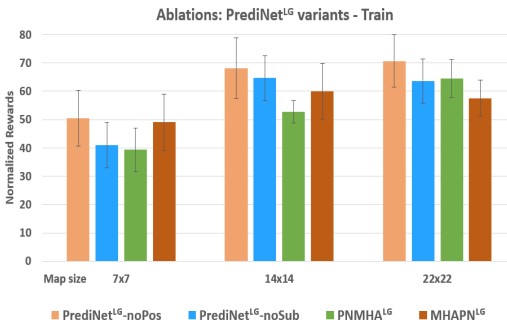 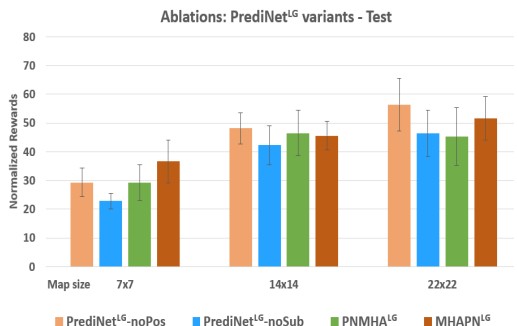

Figure 5: Ablation studies in Minecraft (5 i.r. per variant). PrediNet$_{noSub}^{LG}$ and PrediNet$_{noPos}^{LG}$ are variants of PrediNet$^{LG}$ without the element-wise subtraction and the feature coordinates respectively. PNMHA$^{LG}$ uses an MHA in CM2 while MHAPN$^{LG}$ uses an MHA in CM1.

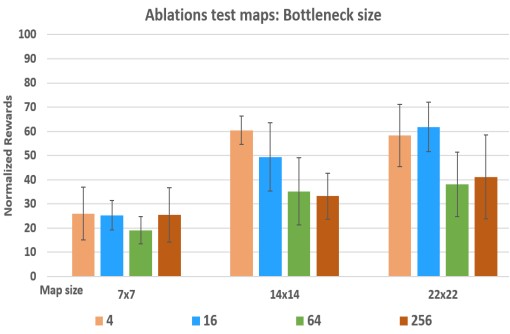 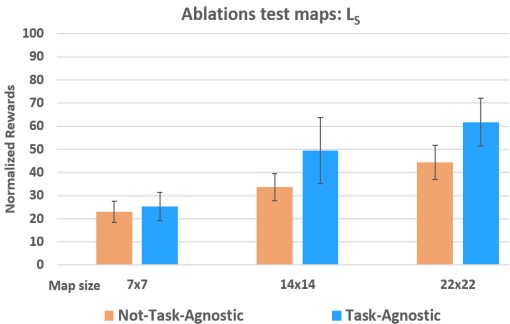

Figure 6: Ablation studies with PrediNet$^{LG}$ in Minecraft (5 i.r. per variant). **Left:** Impact of different bottleneck sizes. **Right:** Impact of having $L_s$ task-agnostic or not.

removal reduces the performance of PrediNet$^{LG}$ to that of an MHA$^{LG}$. None of the networks using an MHA in one of the CMs outperform having a PrediNet in both.

Regarding the outlayer of MHA in MC-14x14, we believe that receiving instructions not preprocessed by a recurrent layer is preventing the MHA to overfit as much as it does in MiniGrid, thus achieving better performance in maps of close-to-distribution size. Still, note that MHA$^{LG}$ outperforms MHA in three out of four OOD results. As for the reason why the standard PrediNet does not outperform the MHA in the same line that PrediNet$^{LG}$ outperforms MHA$^{LG}$, from Shanahan et al. (2020) we note that at the core of the PrediNet information is organised into small pieces that are processed in parallel channels limiting the ways these pieces can interact. This pressures the network to learn representations where each separate piece of information has independent meaning and utility. The result is a representation whose component parts are amenable to recombination. Such feature is highly beneficial for generalization, but supposes that the following recurrent layer needs to combine those pieces while producing a meaningful output for both actor and critic. As highlighted in Santoro et al. (2017), recurrent layers struggle when relating independent pieces of information. The latent-goal configuration alleviates this problem by decoupling the problem of goal abstraction and state representation into separated channels. Consequently, a latent-goal architecture helps to exploit the full potential of the highly re-usable outputs produced by PrediNets. Thus, if we substitute the element-wise subtraction with a FC layer (as we do with PrediNet$_{noSub}^{LG}$), the output is no longer generating independent pieces of information from the parallel channels, which aligns the representations obtained from the PrediNet with the ones generated by an MHA, that employs a single channel of information.

Last, Fig. 6 provides the results of ablation studies regarding conditions (i) and (ii) from Sec. 4. We observe that, for larger bottleneck sizes, generalization performance drops significantly as the agent no longer relies on the generic representations from $L_s$ since biased information from $L_g$ is no longer constrained. That is, larger bottlenecks do not limit the information passed through $L_g$ biasing the agent towards the objects used in the training tasks. We see a similar degradation in performance when providing information about the task through $L_s$, confirming that the benefits are drawn on the generic

representations from an input that cannot tell on its own which objects are beneficial or dangerous. Additionally, we perform control experiments in Minecraft to verify that our agents are learning compositionally from the training instructions, specially with negation, which previous work (Hill et al., 2020; León et al., 2020) have found particularly challenging. We find evidence that with standard architectures only the MLP and MHA are correctly following the OOD tasks. In the case of architectures inducing latent goals we observe that MLP[LG], MHA[LG] and PrediNet[LG] are able to execute correctly the instructions when applied to OOD objects. Results and further discussion is given in Appendix E.

# 6 RELATED WORK

Applying RL to autonomous agents executing formal specifications has become a thriving area of research. The earliest works combining RL and TL focused on tackling temporally extended goals, naturally described as a non-Markovian reward decision process (NRMDP), by generating an equivalent extended MDP were RL algorithms can be applied (Bacchus et al., 1996). Recent contributions in this line investigate increasingly diverse and expressive languages (Brafman et al., 2018; Camacho et al., 2019) or multi-agent systems (León & Belardinelli, 2020; Hammond et al., 2021). Beyond TL, we find methods such as *reward machines* (a type of finite state machine) (Icarte et al., 2018; Xu et al., 2020; Icarte et al., 2019), or RL-specific formal languages such as SPECTRL (Jothimurugan et al., 2019). Closer to our line of work, Kuo et al. (2020) presents a novel RL framework to follow OOD combinations of known tasks expressed in linear-time temporal logic (LTL) by training multiple networks (one per LTL operator and per object). In a similar line, Araki et al. (2021) introduces a hierarchical reinforcement learning framework aimed to learn policies that are optimal and composable while relying on different neural networks each specialized in one subtask. León et al. (2020) provides evidence that DRL agents relying on a single neural network can learn compositionally from the training instructions when the convolutional layers are designed according to the environment where agents operate. Vaezipoor et al. (2021) tackles a similar problem in non-visual scenarios by pretraining a neural network to learn to interpret LTL instructions in navigation-based environments. Our work advances the state-of-the-art in agents following formal instructions by presenting a novel network architecture that improves the performance of DRL algorithms in OOD environments.

Due to the multi-modular nature of the proposed configuration, our work also has links to modular frameworks. These configurations hold promise in improving generalization through its natural compositionality (Jothimurugan et al., 2019). Examples of modular frameworks include DRL agents that combine general control policies, where a global controller is able to combine with different lower-level modules in series of locomotion tasks (Huang et al., 2020). Closer to this work Karkus et al. (2020) propose a framework that makes use of a perception and a planner networks to selects the most suitable human-designed instruction for each time step to be forwarded to a controller network. This differs from our proposed framework, where latent goals are sparsely communicated and learned from scratch without using any predefined human tasks as embedding of the goal.

# 7 CONCLUSIONS

By relying on the assumption that human instructions can be separated from the sensory input, we have presented a novel deep learning configuration that induces situated agents to generate latent representations of their current goal. Particularly, we have seen that deep learning can draw on architectures where output layers take decisions based on both task-agnostic representations of the environment's state and the outputs of a channel that processes observations and human instructions together, we call those outputs *latent goals*. This configuration yields stronger OOD generalization as long as the latent-goal is forced through a small-enough information bottleneck to the decision-making layers. We have provided evidence of the strong potential this configuration has in multiple networks and settings. We believe our findings may have a broad impact within the communities working with DRL, OOD generalization and formal methods. Further, this work poses interesting questions about task abstraction and generalization. It also remains open how well DRL agents can generalize when the symbolic part may not provide reliable feedback on task progression. We think that addressing these questions, among many others, will push the boundaries of applicability in RL.

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

---

**Algorithm 1** Symbolic Module (SM)

---

1: **Input:** Instruction $T$
2: Generate the accepted sequences of tasks $\mathcal{K} \leftarrow \mathcal{E}(T)$
3: Retrieve the current observation: $z$
4: Get the true proposition: $p \leftarrow \mathcal{L}_I(z)$
5: Get the first task: $\alpha \leftarrow \mathcal{P}(\mathcal{K}, p)$
6: **repeat**
7:     Get the next action: $a \leftarrow$ Neural Module (NM)$(z, \alpha)$
8:     Execute $a$, which updates $z$
9:     Get the new true proposition: $p \leftarrow \mathcal{L}_I(z)$
10:     Provide the reward: NM $\leftarrow R(p)$
11:     **if** $p == p_\alpha$ **then**
12:         Update $\mathcal{K}, \alpha \leftarrow \mathcal{P}(\mathcal{K}, p)$
13:     **end if**
14: **until** $\mathcal{K} == \emptyset$

---

## A  THE SYMBOLIC MODULE

In this section we include the pseudo-code of the SM and the internal extractor ($\mathcal{E}$) and progression ($\mathcal{P}$) functions, whose procedures are explained in Sec. 3.2 of the main text.

Algorithm 1 shows the general functioning of the SM and its iterations with the NM. Within the SM, Algorithm 2 details the Extractor $\mathcal{E}$. Given the complex instruction $T$, $\mathcal{E}$ generates a list $\mathcal{K}$ with all the possible sequences of tasks $\alpha$ that satisfy $T$. Note that $\alpha$ are the tasks that the neural module (NM) is expected to learn/solve compositionally. Algorithm 3 refers to the Progression function $\mathcal{P}$, which updates $\mathcal{K}$ according to the true evaluation $p_\alpha$ given by the internal labelling function $\mathcal{L}_I$, that indicates to the SM how the previous task has been solved, and selects the next task $\alpha'$.

## B  ENVIRONMENT DETAILS

Below expand the description of the benchmarks introduced in the main text:

**Minecraft-inpired.** In this environment agents have four actions: $Up, Down, Left, Right$ to move one square from their current position, according to the given direction. Specifically, we use a configuration where maps are procedurally generated by randomly selecting populations of objects, their placement and agent's starting position, and where each tile within has a resolution of 9x9 gray (1 channel) pixels. We generate 50 objects split into various sets. Each object is represented by a matrix of 9x9 values. These values were procedurally generated with pseudo-aleatory numbers (we used a fixed seed). The global set of objects is referred to as $\mathcal{X}$, where the total number of objects is $|\mathcal{X}| = 55$. The set is partitioned into pretraining set $|\mathcal{X}_1| = 35$ , training set $|\mathcal{X}_2| = 20$ and a test set $|\mathcal{X}_3| = 20$, where $\mathcal{X} = \mathcal{X}_1 \cup \mathcal{X}_3$, $\mathcal{X}_2 \subset \mathcal{X}_1$ and $\mathcal{X}_1 \cap \mathcal{X}_3 = \emptyset$.

**MiniGrid.** In this environment there are three possible actions: $Move\ Straight,\ Turn\ Left,$ $Turn\ Right.$ Here objects are formed from compositions of colors and shapes. We expand the number of shapes and colors from vanilla MiniGrid to have eleven colors and eight shapes. We refer to the total set of colors and shapes as $C$ and $F$ respectively. We also divide the global set of instructions in four categories as follows:

- Reachability atomic goals, which have the form "$True\ U + p$".
- Negative reachability goals, e.g., "$True\ U - p_1 \vee - p_2$" or "$True\ U - p$".
- Positive instructions such as "$+ p_1\ U + p_2$" or "$+ p_1 \vee + p_2\ U + p_3 \vee + p_4$".
- Negative conditions, e.g., "$- p_1\ U + p_2 \vee + p_3$".

With atomic reachability goals we use sets $C_1$ and $F_1$ while training and $C_2$ and $F_2$ for testing, where $|C_1| = 8$, $|F_1| = 6$, $C = C_1 \cup C_2$, $C_1 \cap C_2 = \emptyset$, $F = F_1 \cup F_2$ and $F_1 \cap F_2 = \emptyset$. The remaining categories of instructions use $C_3, F_3$ in training and $C_4, F_4$ for testing, where $|C_3| = 8$, $|F_3| = 6$,

---

**Algorithm 2** Extractor function, obtains the list of all the possible sequences of tasks

---

1: **Function** $\mathcal{E}$, **Input:** $T$
2: Initialize the list for the sequences of tasks $\mathcal{K}$
3: **for** each atomic task $\alpha \in T$ **do**
4:    **if** $\alpha$ is atomic positive or negative **then**
5:       **for all** $Seq \in \mathcal{K}$ **do**
6:          $Seq.\text{append}(\alpha)$
7:       **end for**
8:    **else**
9:       // There are non-deterministic choices
10:       Initialize choice list: $CL$
11:       $LK \leftarrow length(\mathcal{K})$
12:       **for all** $Seq \in \mathcal{K}$ **do**
13:          **for all** choice $\in \alpha$ **do**
14:             $CL.\text{append}(\text{choice})$
15:             Generate a clone per choice $Seq' \leftarrow Seq$
16:             $\mathcal{K}.\text{append}(Seq')$
17:          **end for**
18:       **end for**
19:       Initialize counter $c \leftarrow -1$
20:       **for** $i$ in **range**$(length(\mathcal{K}))$ **do**
21:          **if** $i\% LK == 0$ **then**
22:             $c+ = 1$
23:          **end if**
24:          We append a different choice to each sequence cloned $\mathcal{K}[i].\text{append}(CL[c])$
25:       **end for**
26:    **end if**
27: **end for**
28: **return** $\mathcal{K}$

---

$C = C_3 \cup C_4$, $C_3 \cap C_4 = \emptyset$, $F = F_3 \cup F_4$ and $F_3 \cap F_4 = \emptyset$. Also, $C_2 \subset C_3$, $C_4 \subset C_1$, $F_2 \subset F_3$ and $F_4 \subset F_1$. Intuitively, we use reachability goals to ground some shapes and color for our agents, and the rest of the instructions for the remaining elements. Then we test the agents with the unseen colors and shapes for each category.

**Task generation.** We procedurally generate tasks as detailed in Algorithm 4. Where task type ($\alpha_{type}$) refers to one of the four categories of instructions as detailed in MiniGrid paragraph in Appendix B. Note that in Minecraft all task types have the same population of training objects since all the test objects are zero-shot.

**Experiment Architecture.** All the agents in this work use a pretrained encoder. While pretraining we use only instructions of the form $\diamond l = trueUl$, with objects from $\mathcal{X}_1$ in Minecraft, or $C_1$ and $F_1$ in MiniGrid. This pretraining stage helps the encoder to differentiate objects within the environment. Once pretrained, the encoder remains frozen to prevent overfitting. The training stage is 70M timesteps in Minecraft or 50M in MiniGrid, with maps of training size and populated with objects from the training sets. For testing, all the training parameters are frozen and the agents are evaluated in sets of 500 maps populated with OOD objects from the test sets. At the beginning of each episode, a task $\alpha$ is procedurally generated (e.g., $\alpha_1$ or $\alpha_2$ above). Train and test tasks are generated with objects from the corresponding set. In Minecraft, tasks include zero-shot (i.e., unseen) objects. Once we selected a task, a new map is generated where the agent is placed in an aleatory position, some or all the goal objects are randomly placed and some or all the objects for the constraint are also randomly placed. Additional objects from the corresponding set are also included in the map as distractions.

---

**Algorithm 3** Progression function, returns the next task to be solved.

---

1: **Function** $\mathcal{P}$**, Input:**$\mathcal{K}, p_\alpha$
2: **if** $p_\alpha \neq \emptyset$ **then**
3:     **for all** $Seq \in \mathcal{K}$ **do**
4:         **if** $p_\alpha$ fulfills $Seq[0]$ **then**
5:             $Seq$.pop($Seq[0]$)
6:         **else**
7:             $\mathcal{K}$.pop($Seq$)
8:         **end if**
9:     **end for**
10: **end if**
11: $\alpha' \leftarrow \emptyset$
12: **if** $\mathcal{K} == \emptyset$ **then**
13:     **return** $\alpha'$
14: **else**
15:     Select the head of the first sequence: $\alpha' \leftarrow \mathcal{K}[0][0]$
16:     **for all** $Seq \in \mathcal{K}$ **do**
17:         // look for a non-deterministic choice
18:         **if** $\alpha'$ & $Seq[0]$ can be combined **then**
19:             $\alpha' \leftarrow \alpha' \cup Seq[0]$
20:         **end if**
21:     **end for**
22: **end if**
23: **return** $\alpha'$

---

**Algorithm 4** Task generator

---

1: **Input:** is_test
2: Randomly select task type $\alpha_{type}$
3: **if** is_test **then**
4:     Pop $\leftarrow$ Collection of test objects for $\alpha_{type}$
5: **else**
6:     Pop $\leftarrow$ Collection of train objects for $\alpha_{type}$
7: **end if**
8: Randomly select number of goals $n_g$ in range $[1, 2]$
9: **if** $\alpha_{type} \mathrel{!}= 1$ **then**
10:     Defaults number of safety constrains: $n_c = 0$
11: **else**
12:     Defaults number of safety constrains: $n_c = 1$
13: **end if**
14: **if** $\alpha_{type} == 3$ **then**
15:     Randomly select $n_c$ in range $[1, 2]$
16: **end if**
17: Form an instruction $I$ with $n_g$ goals and $n_c$ safety constrains by selecting different objects from Pop for every constrain and goal.
18: **return** $I$

---

## C   Empirical detail

Here we include further details about the empirical evaluation. The repository at `https://github.com/bgLeon/Latent-Goal-Architectures` provides the code used to perform our experiments.

### C.1   Hardware

For training we use various computing clusters with GPUs such as Nvidia Tesla K80, and GTX 1080. We employ Intel Xeon CPUs and 7700k and consume 5 GB of RAM per every three independent

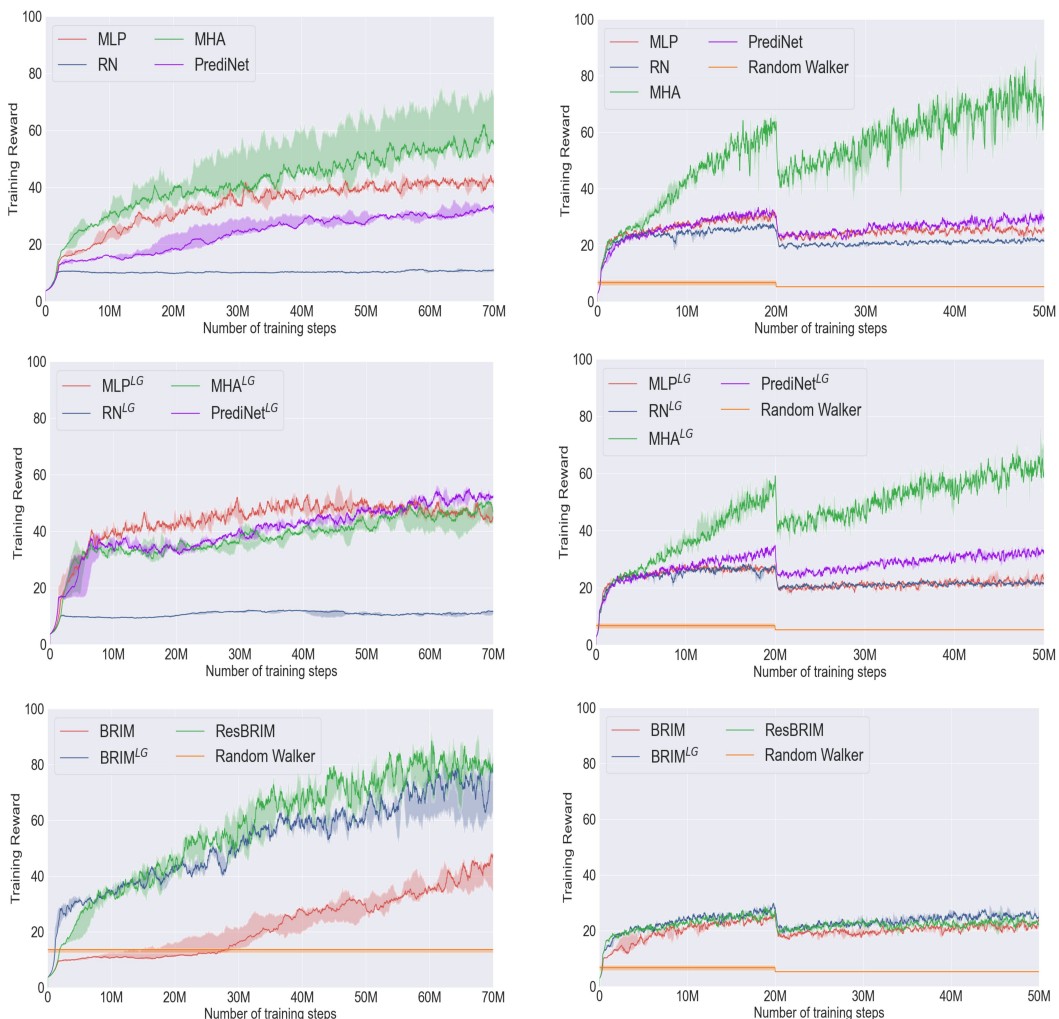

Figure 7: Training plots of the networks studied in Sec 5. Continuous lines correspond to the $50^{th}$ percentiles while the shadowed areas are the $25^{th}$ and $75^{th}$ percentiles. Note that good train performances do not yield to similarly good test results as detailed in the main text. **Left:** Minecraft training plots. **Right:** MiniGrid plots. Note that the general performance drop at 20M in this setting is motivated by the higher chance of generating larger training maps from that point on.

runs working in parallel. Running concurrently, each experiment typically takes 3 days to train, 4 days in the case of latent-goal configurations.

### C.2 RL HYPERPARAMETERS

All of our agents are trained with A2C. Specifically, we use a discount factor $\gamma = 0.99$, a value loss weight of $0.5$ and a batch size of $512$. We work with a scheduled learning rate using a starting value of $8e^{-5}$, $6e^{-5}$ at the step 30 M, and $4e^{-5}$ at 55 M. In the case of BRIMs, which are significantly larger networks, we use a constant learning rate of $3e^{-5}$. These values offered the best results when testing after a grid-search in a range values between $1e^{-5}$ and $1e^{-3}$. The entropy-exploration term was fixed to $H = 1e^{-3}$ after testing with values in the range $[1e^{-5}, 1e^{-2}]$.

### C.3 NETWORKS

Our models use 3-layer convolutional networks in both settings. Particularly, in Minecraft we use a kernel of 3x3 in the first and second layers and 1x1 in the third one. The number of channels in

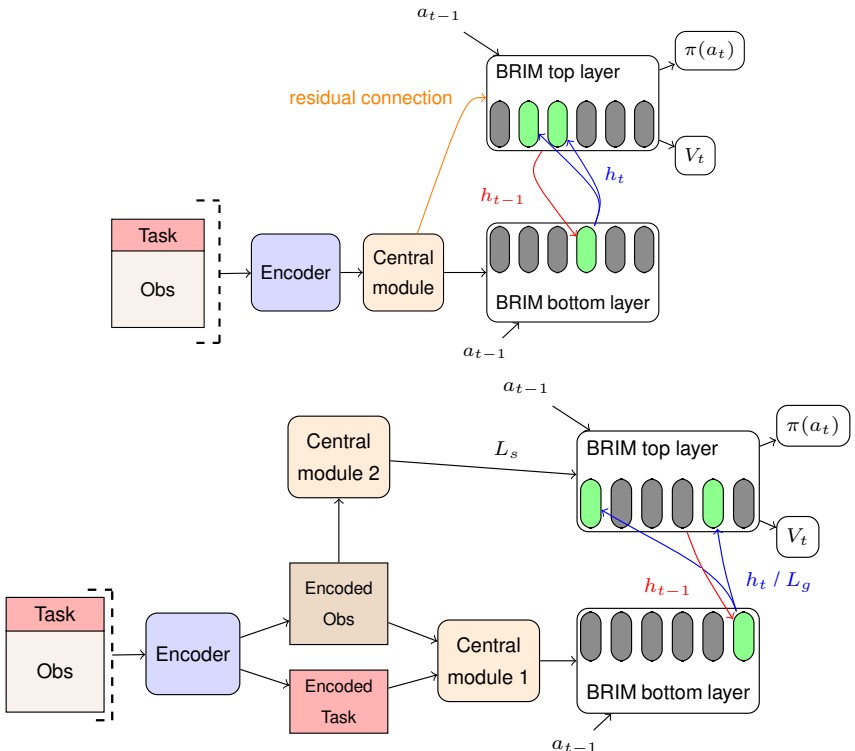

Figure 8: **Top:** vanilla BRIM and ResBRIM (without and with the residual connection respectively). Green cells within the BRIM layer refer to activated modules while gray cells are deactivated. The bottom layer receives as input the output of the central module and the hidden state from the upper layer in the previous time step ($h'_{t-1}$). The upper layer receives the output from the bottom layer ($h_t$). Only strong connections (the ones between activated modules) are shown. In the case of ResBRIM the upper layer also receives the output of the Central module. **Bottom:** BRIM$^{LG}$. The architecture is similar to the ResBRIM but the top layer receives as additional input an embedding from the observation exclusively ($L_s$). Consequently, the top layer has only access to the current goal through the lower layers of the hierarchy.

each layer are 8, 16 and 1 while the strides are 3, 3 and 1 respectively. Note that this encoder has been designed bearing in mind the resolution of the tiles within so that the latent representations (i.e., the outputs of the encoder) of the tasks are independent from the latent representations of the original observation $z$. This inductive bias is exploited in the latent-goal architectures. In MiniGrid all kernels and strides are of size 2. The number of channels are 16, 32 and 32 respectively. Note that encoders are pretrained and then frozen, we found that doing this improved the performance of all agents. With MiniGrid's text instructions we use a bidirectional recurrent layer of size 32. In multi-layer configurations the other two recurrent layers are LSTMs of size 128. The information bottleneck is a FC layer of size 16.

For the central modules in the multilayered architectures, we use the best hyperparameters from a previous ablation study in Shanahan et al. (2020). Particularly, in the case of the MLP we use a single fully-connected layer with 640 units. The RN has a central hidden size of 256 units and an output size of 640 units. The MHA employs 32 head with a key size of 16 and value size of 20. Last, the PrediNet has 32 heads, with key and relation sizes of 16. The latent-goal variants use a FC layer of size 16 for the information bottleneck. Regarding BRIMs we use the hyperparameters recommended for RL in Mittal et al. (2020). Specifically, we use 2 hierarchical BRIM layers, with 6 modules of 50 LSTM units each. Four modules ($k$) are active at a given time step (top-$k$ mechanism). Figure 8 illustrates the different BRIM configurations that we evaluate in Sec. 5 in the main text when working with visual instructions, i.e., the Minecraft setting.

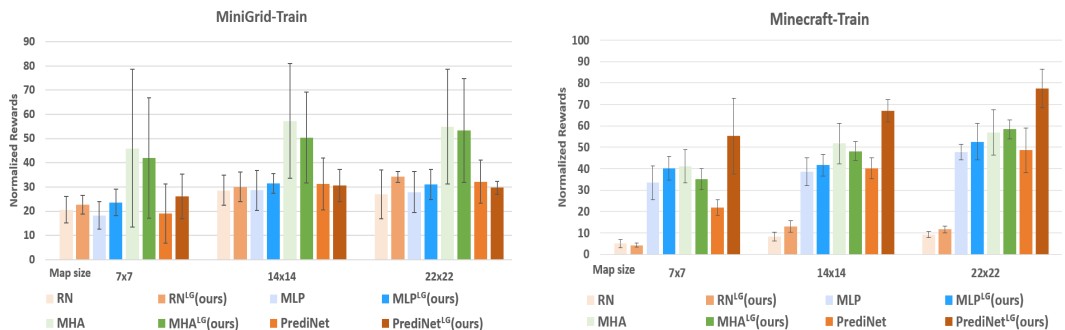

Figure 9: Results with training instructions in 500 maps (per size) of different dimensions. Sizes 14 and 22 are OOD.

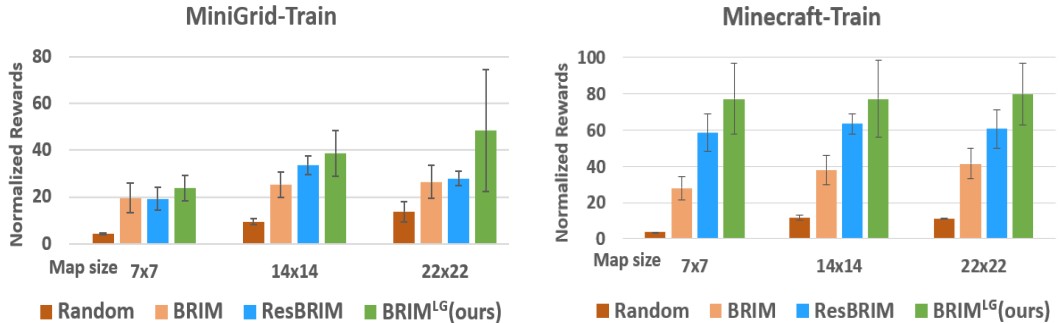

Figure 10: Results of 5 i.r. of vanilla BRIM (BRIM), BRIM with residual connections (ResBRIM), and with LGs (BRIM$^{LG}$).

# D    ADDITIONAL EMPIRICAL ANALYSIS

In this section we first include the training results and tables with the specific values used to generate the bar-charts. Then, we detail the results of the statistical analysis (a two-way ANOVA) of the results with multi-layered networks. Last, we provide the training plots of the experiments in the main document.

## D.1    TRAINING PERFORMANCE AND DETAILED VALUES

Figs. 9-11 illustrate the training results of the experiments in the main section. Tables 1-4 provide the specifics values in a table format. From the training results, we observe that latent-goal configurations

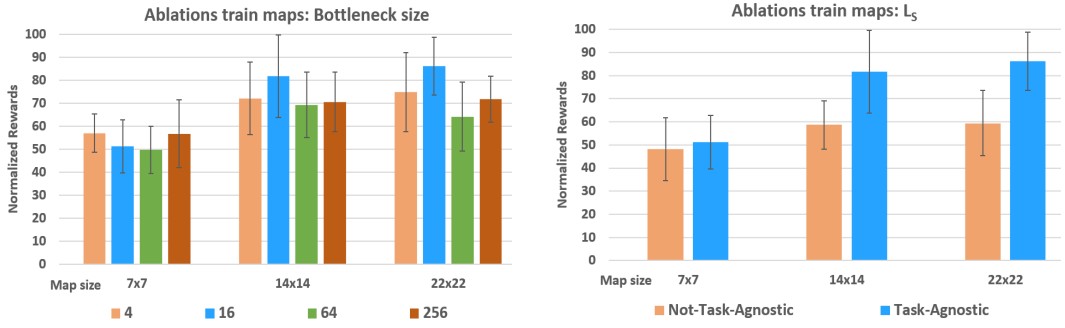

Figure 11: Ablation studies with PrediNet$^{LG}$ in Minecraft (5 i.r. per variant). **Left:** Impact of different bottleneck sizes. **Right:** Impact of having $L_s$ task-agnostic or not.

Table 1: Results with either training (train) or OOD instructions (test) in 500 maps (per size) of different dimensions. Note that sizes 14 and 22 are OOD for all the agents. MC refers Minecraft whereas MG to MiniGrid. Results show the average reward and standard deviation from 10 independent runs (i.r.). Values are normalized so that 100 refers to the highest performance achieved by the best run globally in maps of the given size and benchmark. Best average rewards in each setting and size is bolded.

| Map-size | MLP | | RN | | MHA | | PrediNet | |
|---|---|---|---|---|---|---|---|---|
| | Train | Test | Train | Test | Train | Test | Train | Test |
| MG-7x7 | 18.2(5.7) | 14.1(3.6) | 20.6(5.4) | 18.7 (6.4) | **45.9**(26.6) | 9.2(3.7) | 19.1(12.2) | 15.9 (7.3) |
| MC-7x7 | 33.4(7.9) | 21.1(3.2) | 5(1.9) | 3.2 (1.6) | 41.1(7.7) | **36.6**(4.3) | 21.8(3.7) | 16.1(3.5) |
| MG-14x14 | 28.6(8.2) | 21.1(5.7) | 28.6(6.2) | 27.5(6.0) | **57.3**(17.8) | 21.9 (4.7) | 31.2(10.7) | 25.6(7.3) |
| MC-14x14 | 38.7(6.5) | 32.0(4.5) | 8.3(2) | 7.9(2.4) | 51.7(9.3) | 45.8 (7.6) | 40.2(5) | 31.9(6.2) |
| MG-22x22 | 27.8(8.5) | 21.6(4.3) | 26.9(10.1) | 24.3(5.1) | **54.9**(23.6) | 22.7(6.8) | 32.1(8.9) | 24.1(8.0) |
| MC-22x22 | 47.8(3.6) | 37.8(3.8) | 9.2(1.5) | 8.3(1.8) | 56.9(10.7) | 49.2(5) | 48.7(10.3) | 38(8.5) |

| Map-size | MLP$^{LG}$ (ours) | | RN$^{LG}$ (ours) | | MHA$^{LG}$ (ours) | | PrediNet$^{LG}$ (ours) | |
|---|---|---|---|---|---|---|---|---|
| | Train | Test | Train | Test | Train | Test | Train | Test |
| MG-7x7 | 23.6(5.5) | 21.6(2.6) | 22.6(3.3) | 21.8 (2.8) | 42.0(24.9) | 15.4(3.5) | 32.1(9.3) | **26.1**(5.8) |
| MC-7x7 | 40.2(5.7) | 24.7(2.6) | 4.4(1) | 4.2 (0.7) | 35.2(5) | 31.6(5.3) | **55.2**(17.6) | 29.5(8.2) |
| MG-14x14 | 31.4(4.2) | 24.6(3.8) | 30.1(6.1) | 29.2(6.9) | 50.4(14.9) | 24.0 (9.4) | 37.0(6.7) | **30.6**(5.5) |
| MC-14x14 | 41.7(5.1) | 35.4(4.2) | 13.1(2.6) | 11.5(2.2) | 48.1(4.4) | 41.8 (4.9) | **67**(5.2) | **51.7**(6.5) |
| MG-22x22 | 31.0(6.3) | 22.5(5.0) | 34.2(2.3) | 25.1(1.5) | 53.3(21.4) | 25.4(8.2) | 39.9(2.7) | **29.7**(7.0) |
| MC-22x22 | 52.6(8.6) | 41.3(7.7) | 11.7(1.5) | 10.7(2.1) | 58.4 (4.5) | 51.5(5.5) | **77.5**(9.1) | **65.3**(6.1) |

Table 2: Results from 5 i.r. of a vanilla BRIM (BRIM), a BRIM with residual connection (ResBRIM) and BRIM with latent goals (BRIM$^{LG}$). Best results for each setting and size are in bold.

| Map-size | BRIM | | ResBRIM | | BRIM$^{LG}$ (ours) | | Random walker |
|---|---|---|---|---|---|---|---|
| | Train | Test | Train | Test | Train | Test | |
| MG-7x7 | 19.6(6.2) | 15.1(4.7) | 19.2(4.8) | 16.7 (3.6) | **23.8**(5.4) | **17.1**(5.3) | 4.3(0.4) |
| MC-7x7 | 27.9(6.3) | 15.6(2.8) | 58.6(10.4) | 15.8 (2.3) | **77.2**(19.5) | **26.4**(7.0) | 3.7(0.1) |
| MG-14x14 | 25.3(5.4) | 19.9(6.2) | 33.7(3.9) | 25.0 (1.4) | **38.8**(9.8) | **26.8**(5.6) | 9.5(1.3) |
| MC-14x14 | 37.9(8.2) | 26.5(6.7) | 63.5(5.5) | 39.6(7.0) | **77.3**(21.3) | **46.8** (8.1) | 11.5(1.4) |
| MG-22x22 | 26.5(7.1) | 20.0 (7.6) | 27.9 (3.0) | 25.3(2.1) | **48.5**(26.0) | **30.3**(6.6) | 13.6(4.2) |
| MC-22x22 | 41.5(8.4) | 29.3(4.1) | 60.6(10.4) | 40.7(8.2) | **79.8**(17.0) | **58.4**(14.8) | 11.1(0.5) |

can improve the performance also the with training instructions and that this improvement becomes more noticeable as environments become larger in size, i.e., as we get further from the training distribution of environments. The fact that the effects of latent-goal networks are most noticeable with unseen instructions aligns with the conclusions drawn from the main document, where we stated that this architectures are specially conceived to strength agents' ability to generalize to new tasks.

## D.2 STATISTICAL ANALYSIS

To confirm that the standard error of the different conditions do not overlap each other, we perform a two-way analysis of variance (two-way ANOVA) with the independent variables being the neural networks used in the central modules and the architecture configuration, i.e., latent goal (ours) or standard configuration. The 2-way ANOVA is done across the maps of different size of Minecraft and MiniGrid respectively. Note that to evaluate the different hypotheses, we need to check whether the F value is greater than its corresponding F-critic or not, and to confirm that the P-value is smaller than Alpha.

Table 5 shows the results of the ANOVA analysis with the results from Minecraft. We see that it confirms that both the architecture configuration and the neural network have a significant impact of the performance of the agent. Noticeably, the ANOVA results show that there is not a significant interaction between the two variables, meaning that the neural network is not relevant in the general improvement in performance that latent goals grant in this setting. Still, note that this is analysis of the statistical impact averaged across all the networks. As studied in Sec. 5.1 the latent goals do have a different impact with the PrediNet than with other networks.

Table 3: Ablation studies in Minecraft (5 i.r. per variant). $PrediNet^{LG}_{noSub}$ and $PrediNet^{LG}_{noPos}$ are variants of $PrediNet^{LG}$ without the element-wise subtraction and the feature coordinates respectively. $PNMHA^{LG}$ uses an MHA in CM2 while $MHAPN^{LG}$ uses an MHA in CM1. Best results are in bold

| Ablations | 7x7 | | 14x14 | | 22x22 | |
|---|---|---|---|---|---|---|
| **$PrediNet^{LG}$ variants** | Train | Test | Train | Test | Train | Test |
| $PrediNet^{LG}_{noSub}$ | 41 (8) | 22.8 (2.8) | 64.7 (7.9) | 42.3 (6.9) | 63.7 (7.8) | 46.4 (8.1) |
| $PrediNet^{LG}_{noPos}$ | **50.6** (9.9) | 29.3 (5) | **68.2** (10.8) | **48.1** (5.4) | **70.6** (9.3) | **56.4** (9.2) |
| $PNMHA^{LG}$ | 39.3 (7.6) | 29.2 (6.2) | 52.8 (3.9) | 46.5 (7.9) | 64.6 (6.7) | 45.2 (10) |
| $MHAPN^{LG}$ | 49 (10) | **36.6** (7.5) | 59.9 (9.8) | 45.6 (4.9) | 57.4 (6.4) | 51.6 (7.6) |

Table 4: Ablation studies with $PrediNet^{LG}$ in Minecraft (5 i.r. per variant). In the first set we explore the impact of different bottleneck sizes. In the second we contrast having $L_s$ task-agnostic or not. For every ablation set, best results are bolded.

| Ablations | 7x7 | | 14x14 | | 22x22 | |
|---|---|---|---|---|---|---|
| **Bottleneck size** | Train | Test | Train | Test | Train | Test |
| 4 | **57.0** (8.4) | **26.0** (10.9) | 72.0 (15.7) | **60.4** (5.9) | 74.8 (17.2) | 58.3 (12.9) |
| 16 | 51.2 (11.6) | 25.3 (6.1) | **81.7** (17.9) | 49.4 (14.2) | **86.1** (12.6) | **61.8** (10.3) |
| 64 | 49.8 (10.2) | 19.1 (5.7) | 69.3 (14.3) | 35.2 (13.9) | 64.2 (15.1) | 38.1 (13.3) |
| 256 | 56.7 (14.7) | 25.4 (11.3) | 70.5 (13.0) | 33.2 (9.5) | 71.7 (10.1) | 41.2 (17.3) |
| **$L_s$ data type** | | | | | | |
| Task-agnostic | **51.2** (11.6) | **25.3** (5.4) | **81.7** (17.9) | **49.4** (12.6) | **86.1** (12.6) | **61.8** (10.3) |
| Not Task-agnostic | 48.1 (13.6) | 23.0 (4.6) | 58.7 (10.4) | 33.7 (5.8) | 59.4 (14.1) | 44.4 (7.3) |

Table 6 shows the results of the ANOVA test with MiniGrid. Here we see that neural networks have a significant impact in the training performance of the agent but that is not the case with the architecture, i.e., using latent goals does not have a significant impact in the global training performance. Nevertheless, we see that this changes with unseen instructions (test) where the latent goals have a stronger impact in performance than the neural network, being both statistically relevant. Last, from the interactions' results we see that both in training and test the interaction of the two variables have a significant impact in the final performance, i.e., the effect in performance that latent goals have is strongly dependant on the neural network and vice versa.

Regarding the differences in how network and architecture impact in Minecraft and MiniGrid, we believe that these are motivated by the additional difficulty of MiniGrid observation and action settings. Specifically, MiniGrid only allows agents to move forward or turn around, and agents can only observe what is in front of them. This requires agents to rely more on their memory than they do in Minecraft, where they can observe the objects around them in any direction and have actions to move in four different directions. This point also motivated the need of using an schedule in the introduction of larger training maps in MiniGrid, as detailed in Appendix D.3

## D.3 TRAINING PLOTS

Figure 7 shows the training plots of the standard (top), latent-goal (middle) and BRIM (bottom) configurations in Minecraft (left) and MiniGrid (right). In the later it can be appreciated a general decrease of performance the 20M mark. This is because we noticed that in MiniGrid, where the navigational features force the agent to further rely on memory, we noticed that agents struggled when learning with the default configuration of random sizes with $n = [7 - 10]$. To alleviate this, through the first 20 M steps we give a higher chance (70%) to maps of size 7, to help agents to learn from the instructions before navigating through larger maps. Note that good training performance, as shown in these training plots, does not necessarily mean good OOD performance. This is highlighted in Table 1.

Table 5: Two-way ANOVA test with Alpha= 0.05 in Minecraft. Neural network refers to the impact of the different neural networks in the central modules, whereas architecture configuration refers to the impact of whether using a latent goal or a standard configuration.

| **Train** | $F$ | $P-value$ | $F-critic$ |
|---|---|---|---|
| Architecture configuration | 43.56 | 2.74exp$-10$ | 3.88 |
| Neural Network | 237.31 | 2.13exp$-70$ | 2.64 |
| Interactions | 31.27 | 5.16exp$-17$ | 2.64 |
| **Test** | $F$ | $P-value$ | $F-critic$ |
| Architecture configuration | 23.63 | 2.15exp$-6$ | 3.88 |
| Neural Network | 166.75 | 1.26exp$-57$ | 2.64 |
| Interactions | 16.26 | 1.26exp$-09$ | 2.64 |

Table 6: Two-way ANOVA test with Alpha $= 0.05$ in MiniGrid.

| **Train** | $F$ | $P-value$ | $F-critic$ |
|---|---|---|---|
| Architecture configuration | 1.37 | 0.24 | 3.93 |
| Neural Network | 39.57 | 1.63exp$-17$ | 2.69 |
| Interactions | 1.86 | 0.14 | 2.69 |
| **Test** | $F$ | $P-value$ | $F-critic$ |
| Architecture configuration | 8.57 | 0.004 | 3.93 |
| Neural Network | 5.51 | 0.001 | 2.69 |
| Interactions | 0.53 | 0.6 | 2.69 |

## E    Compositional learning

Here we assess whether our agents are learning compositionally from SATTL instructions and check if they are really following the given safety constraints when applied to OOD objects or if they are just learning to reach the goal $g_\alpha$. We focus our study on their performance with negated constraints, since previous research highlighted as the most challenging type of instructions Hill et al. (2020); León et al. (2020). Compositional learning Lake (2019) (also known as systematic learning Hill et al. (2020)) refers to the ability of understanding and producing novel utterances by combining already known primitives Chomsky & Lightfoot (2002). In our context, an agent that is learning compositionally should be able to solve the instruction $-c\,U + p$ if the primitives $c$ and $p$ are known and it already learnt to solve $-c'U + p'$.

Table 7 shows the results from a control experiment where we track the performance of the agents according to tasks of the form $-c\,U + p$ when giving reliable, partially occluded or deceptive instructions. Specifically, every row shows the test performance of the multi-layered networks trained in Sec. 5 when providing rewards according to the instruction $-c\,U + p$, which intuitively means "avoid $c$ until reaching $p$". The first row shows the performance of the agents when the SM provides *reliable instructions* to the NM, i.e., the instruction provided is the one used to generate rewards. The second row shows the performance under the same setting, but where the extended observation provided to the NM is a reachability goal (true $U + p$). Intuitively, for the second row the agents are given the right goal $p$ but not the safety constraint, i.e., partially occluding information of the real task. The third row shows the performance when the SM gives deceptive information abut the safety constrain ($+cU + p$), i.e., the SM is "telling" the NM to go through the objects that actually should be avoided. Agents learning systematically should show worse performance with partially occluded instructions than with the reliable ones, but still better than a random walker. Additionally, the worst performance should come when provided deceptive instructions.

From Table 7 we see that all the variants using the MHA and MLP modules follow the rule $1^{st}$ row $> 2^{nd}$ row $>$ random $> 3^{rd}$ (compositional rule, or c.r. for short). Notably, the best performance comes from a variant that does not use relational networks nor attention (MLP$^{LG}$). Still, this does not imply that MLPs are better suited for negation since agents were trained in a much wider variety of tasks and the MHA, MHA$^{LG}$ and PrediNet$^{LG}$ outperformed the MLP$^{LG}$ in the general evaluation test. In the case of the RN, the c.r. is satisfied but the general low performance and the small difference between the results of the first and the second rows (14 and 13.3 respectively) indicates weak generalization. This is worse with the M-RN and the PrediNet having both equal or better

Table 7: Study on compositional learning with zero-shot objects and instructions. Results show the performance obtained by each network in 200 test maps when rewards are given according to the "real goal", while the symbolic module provides the "given instruction" to the neural module. An agent learning compositionally should have $1^{st}$ row $> 2^{nd}$ row $> 3^{rd}$ row for the values within its column. Also, values lower than random are only acceptable in the third row (deceptive instruction). The mean performance of a random walker is 12.5 independently of the "given instruction".

| | Real goal: $-c\,U + p$ | | | | | | | |
|---|---|---|---|---|---|---|---|---|
| Given instruction | MLP | MLP[LG] | RN | RN[LG] | MHA | MHA[LG] | PrediNet | PrediNet[LG] |
| $-c\,U + p$ | 44.6(10.5) | 70.6(11.9) | 14(1.1) | 13.9 (1) | 55.8(6.1) | 43.8(7.3) | 17.5(1.2) | 43.6(9.3) |
| true $U + p$ | 20.3(5.7) | 18.0(4.6) | 13.3(0.9) | 13.9(1.4) | 16.9(3.0) | 19.9 (6.3) | 19.9(3.4) | 17.5(3.8) |
| $+c\,U + p$ | 8.7(1.6) | 11.0(2.6) | 7.9(2.2) | 5.6(1.2) | 10.8(2.6) | 15.0(3.6) | 8.2(1.8) | 8.4(1.2) |

performance with partially occluded instructions than when receiving the real task as input. Such results imply that these networks have not correctly learnt to generalize negated instructions. This is not the case with the PrediNet[LG], whose results correctly follow the c.r.. In addition, the worse performance of the PrediNet[LG] than the MHA with partially occluded and deceptive instructions suggests that the PrediNet[LG] advantage over the MHA in maps of OOD size ( see Table 1 comes from a better ability of the former to execute safety constraints when compared with the later. Note that the larger the map the harder it is to find $p$; thus the bigger chances of accumulating penalizations due to "violations" of the safety condition.

**Discussion**   To the best of our knowledge, the only two works that include some empirical evidence of emergent compositional learning with negation, i.e., achieving a performance 50% better than chance are Hill et al. (2020); León et al. (2020). However, in those works negated instructions are interpreted as "something different from", e.g., "not $p$" intuitively meant "get something different from $p$". Instead, here we have an interpretation of negated atoms more aligned with propositional and temporal logic, and also to natural language, where "not $p$" intuitively means that $p$ must be false. Hence, we believe this is the first work to show that DRL agents are capable of learning the abstract operator of negation in its classical interpretation and successfully apply it to new utterances.

## F   SATTL AND $\text{LTL}_f$

We stated that SATTL is a fragment of the widely-used $\text{LTL}_f$. The syntax of $\text{LTL}_f$ is defined as follows:

$$\varphi \ ::= \ p \mid \neg\varphi \mid \varphi_1 \wedge \varphi_2 \mid \bigcirc\varphi \mid \varphi_1 U \varphi_2$$

Since both $\text{LTL}_f$ and SATTL are defined ofer finite traces, we can directly introduce the following translations:

$$\varphi \ ::= \ p \mid \neg\varphi \mid \varphi_1 \wedge \varphi_2 \mid \bigcirc\varphi \mid \varphi_1 U \varphi_2$$

**Definition 3.** *Translations $\tau$ from Task Temporal Logic to $\text{LTL}_f$ are defined as follows:*

$$
\begin{aligned}
\tau(\alpha) &= lUl' \\
\tau(T \cup T') &= \tau(T) \vee \tau(T') \\
\tau(T; T') &=
\begin{cases}
lU(l' \wedge \tau(T')) & \text{if } T = \alpha \\
\tau(T_1; (T_2; T')) & \text{if } T = T_1; T_2 \\
\tau((T_1; T') \cup (T_2; T')) & \text{if } T = T_1 \cup T_2
\end{cases}
\end{aligned}
$$

We immediately prove that translation $\tau$ preserve the interpretation of formulae in SATTL.

**Proposition 1.** *Given a model $\mathcal{N}$ and trace $\lambda$, for every formula $T$ in SATTL,*

$$(\mathcal{N}, \lambda) \models T \quad iff \quad (\mathcal{N}, \lambda) \models \tau(T)$$

*Proof.* The proof is by induction on the structure of formula $T$. The base case follows immediately by the semantics of SATTL and $\text{LTL}_f$.

As for the induction step, the case of interest is for formulae of type $T; T'$. In particular, $(\mathcal{N}, \lambda) \models T; T'$ iff for some $0 \leq j < |\lambda|$, $(\mathcal{N}, \lambda[0, j]) \models T$ and $(\mathcal{N}, \lambda[j+1, |\lambda|]) \models T'$. By induction hypothesis, this is equivalent to $(\mathcal{N}, \lambda[0, j]) \models lUl'$ and $(\mathcal{N}, \lambda[j+1, |\lambda|]) \models \tau(T')$ in the case of $T = lUl'$. Finally, this is equivalent to $(\mathcal{N}, \lambda) \models lU(l' \wedge \tau(T'))$. The cases for $T = T_1; T_2$ and $T = T_1 \cup T_2$ are dealt with similarly.

Finally, the case for $T \cup T'$ follows by induction hypothesis and the distributivity of $\vee$. $\qquad\square$

