# OpenReview forum: "In a Nutshell, the Human Asked for This: Latent Goals for Following Temporal Specifications"
_ICLR.cc/2022/Conference — ICLR 2022 Poster_

### Official Review · Reviewer_2S8A · 2021-11-01

**Correctness:** 3
**Technical Novelty And Significance:** 2
**Empirical Novelty And Significance:** 3
**Recommendation:** 6
**Confidence:** 4

**Main Review:**

Strengths:

It seems like a good idea to include environmental information in the reward specification phase, and empirical results do show that this is effective.

Weaknesses:

At present the paper suffers from severe clarity issues, which make it difficult to evaluate. Specifically:

- The abstract doesn't actually explain the task or method, and reads more as a collection of jargon. This needs to be reworked to more clearly specify the problem formulation, goal and proposed solution.

- Contributions need to be made clearer

- Problem formulation: the paper is in need of a clear problem formulation early on, explain exactly what the task is, inputs and outputs, and key steps in the solution, early on. Right now, the background provides an MDP definition that could be repurposed to explain the task, and motivate the key reasons for the proposed framework much earlier on.

- SATTL section: it's unclear what this adds to the paper - right now this reads as a floating intro to SATTL that just breaks the flow of the text. I'd move this to an appendix, or rework to explain only what is required from the perspective of reward specification.

- I only started to understand the paper in section 3.3 - I'd consider moving this much earlier, and motivating the problem using this example.

- It's difficult to map the framework in 3.2 onto the architecture in Fig 1. The method description needs a lot of work.
-Fig 1 and the associated discussion is very confusing. Right now, I'm unsure if the proposed solution only operates in the reward specification phase, and is then used to optimise an actor-critic policy, or if something interesting happens in the latent state of an actor critic network, or if a meta-agent is being trained to predict a zero-shot value function and policy.

Overall, I believe this paper would benefit greatly from being re-read from the perspective of someone not involved in the project, because at present it does not adequately communicate the work completed.

**Summary Of The Paper:**

This paper introduces an neural architecture for a DRL agent tasked with learning to solve temporal logic specifications. Results show the proposed architecture outperforms existing ones when it comes to performance on instruction following in unseen environments.The proposed architecture takes in an observation and a task description, which appears to be transformed into a reward function that is then optimised. The key contribution appears to be around the inclusion of some latent state information about the environment in the reward specification phase, with the motivation that this makes it easier to interpret a human temporal logic specification of a task.

Results appear to be strong, and do show that this results in substantial performance increases in unseen tasks.

Unfortunately though, I found this paper a difficult read and in need of substantial work before it is ready for publication, primarily around the problem framing, method description, discussion and motivation. At present, it lacks the clarity required to best showcase the results, and I cannot confidently say I understand this paper, because it is not explained or presented clearly enough.

**Summary Of The Review:**

I think there is interesting work in here, and a nice story to be told. Unfortunately, the current structure of the paper is not good enough to support this, and the paper is in need of a substantial amount of rework to improve it's clarity.

At present, it lacks the clarity required to best showcase the results, and I cannot confidently say I understand this paper, because it is not explained or presented clearly enough.

==== Post rebuttal comments ====
Thank you for revising this work, I think it is much stronger now after the restructuring and clarification, so have increased my score.

---

> ### Author Response · Authors · 2021-11-18
> **Response to Reviewer 2S8A (1/2)**
>
> We thank the reviewer for their review and detailed comments. Since all the concerns are focused on clarity, we clarify the misinterpretations below and detail the updates and clarifications done in the updated version following those comments. Note that besides the clarifications we detail here, we also made multiple other changes to improve readability and presentation as suggested by other reviewers (such as substituting tables with bar charts and adding a statistical analysis of our results).
>
> > The key contribution appears to be around the inclusion of some latent state information about the environment in the reward specification phase, with the motivation that this makes it easier to interpret a human temporal logic specification of a task.
>
> Please note that the reward specification is completely independent from the neural network architecture. The symbolic and neural modules are independent from each other. The reward function defined in the symbolic module simply generates a reward function according to the current SATTL task, but from the perspective of the DRL algorithm is analogous to any standard reward signal in a POMDP (only that rewards are related to the instruction given).
>
> We updated the name of "Internal Reward" to "Reward" when explaining the symbolic module in Section 3.2 (to avoid further confusion) and included further clarifications while explaining the framework in the updated version.
>
> We also updated the introductions of Sections 3, 3.2 and old Section 3.3 (now Section 4).
>
> We made Section 3.3 an independent section (Section 4) to better highlight that this is our main contribution. As further explained there, the key element in our architecture is that the "decision making" layers are receiving information from two channels: first $L_s$, which is a task-agnostic representation of the environment, second $L_g$, that process both observation an human instruction and that is passed through an "information bottleneck" limiting the information that comes from $L_g$ (and forcing the decision-making layers to use the unbiased representations from $L_s$).
>
> Additionally, we included further clarifications in Section 4 (figure 1 and its explanation).
>
> > The abstract doesn't actually explain the task or method, and reads more as a collection of jargon. This needs to be reworked to more clearly specify the problem formulation, goal and proposed solution.
>
> Thank you for the pointer, we rewrote the abstract to better specify what the work does.
>
> > Contributions need to be made clearer
>
> We updated the contributions paragraph and presented them as a bulletized list to improve readability
>
> > Problem formulation: the paper is in need of a clear problem formulation early on, explain exactly what the task is, inputs and outputs, and key steps in the solution, early on. Right now, the background provides an MDP definition that could be repurposed to explain the task, and motivate the key reasons for the proposed framework much earlier on.
>
> We believe that the updated abstract, Section 1 and introductions of Sections 3,3.1, 3.2 and 4 tackled this. Introducing formally the POMDP is necessary since we repeatedly use terms such as states, actions, observations and rewards through the work.
>
> > SATTL section: it's unclear what this adds to the paper - right now ...
>
> This section (3.1) formally states syntax and semantics of the formal language that we use to generate instructions and evaluate the agents. Intuitively, delimits formally the expressiveness of the tasks we want our agents to learn compositionally from. It also includes  examples of tasks generated in the experiments.
>
> We clarified this point in the updated introduction of  Section 3.1
>
> > I only started to understand the paper in section 3.3 - I'd consider moving this much earlier, and motivating the problem using this example.
>
> We now give this example in Section 1
>
> > It's difficult to map the framework in 3.2 onto the architecture in Fig 1.
>
> This is related to a misinterpretation we mentioned above, the architectures in Figure 1 depict the different neural networks architectures that we test within the neural module of the framework explained in section 3.2. We believe the clarifications we mentioned before will help with that, but to further avoid confusion we changed the old title of figure 1 (which was "Model Architectures") to "Neural Network Architectures".
>
> We respond to the last two points in the comment below

---

> > ### Author Response · Authors · 2021-11-18
> > **Response to Reviewer 2S8A (2/2)**
> >
> > > Fig 1 and the associated discussion is very confusing.
> >
> > We believe this is answered in our first response above. We updated Section 4 and Figure 1, highlighting the decision modules that we refer to in the updated description.
> >
> > > Overall, I believe this paper would benefit greatly from being re-read from the perspective of someone not involved in the project
> >
> > The submitted version had been re-read by someone outside the project, but what is seen as "clear" many times changes from reader to reader (as can be seen in the different reviews). That's why we are sincerely very grateful for all your specific feedback, we hope that the updates help to raise your confidence in our work. We are happy to incorporate any additional suggestions you might have to further improve it.

---

> > > ### Author Response · Authors · 2021-11-22
> > > **Follow up**
> > >
> > > Please let us know if the reviewer is satisfied with our response and updated draft, note that the response period for authors ends today. We are happy to address any further questions.

---

> > > > ### Author Response · Authors · 2021-11-29
> > > > **Thank you**
> > > >
> > > > We would like to thank the reviewer for the time spent on our paper again. We are glad to hear that the modifications we incorporated to the paper made it much stronger and easier to read. The feedback we received was definitely instrumental to achieve this.
> > > >
> > > > Thanks for updating the score.
> > > >
> > > > Best regards,

---

### Official Review · Reviewer_FvKn · 2021-11-03

**Correctness:** 2
**Technical Novelty And Significance:** 2
**Empirical Novelty And Significance:** 1
**Recommendation:** 3
**Confidence:** 4

**Main Review:**

Section 3.2: This approach, especially with the definition of the symbolic module, seems to bypass many of the challenging problems in instruction-following: sparse rewards (e.g., rewards only available at the end of the episode), with no explicit progress-monitoring provided, noisy instructions and environment, no basis for task-specific reward-shaping, goals are not necessarily conditioned on the agent’s current state (i.e., leading to issues with initial heading, localisation, etc.). The SM provides a lot of privileged information, through its labeling function and comprehensive reward function.

Section 3.2: How is ambiguity handled — i.e., when there exists multiple ways that T can be entailed?

Section 3.3: The manuscript states "Consequently, it can be beneficial for agents to compute both human instructions and sensory inputs in a dedicated channel, generating a latent representation of the current goal. We call this type of deep learning model a latent-goal architecture." By itself, this is not novel: there are a plethora of works in learning robot skills, and they have dramatically less supervision.

Section 4: Table 1: Train/Test results should be compared separately. Bolding schemes (to highlight the best-performing model) change, from table to table; this should be made consistent.

Section 4: The manuscript, which pursues so-called 'latent goal architectures' for following formal instructions, somehow makes reference to neither the robot skills literature nor the robot instruction-following literature. Additional experiments and comparisons are needed.

Throughout: The manuscript relies on the assumption that "human instructions can be separated from the sensory input". This is a strong and unrealistic assumption, in practice. What need to we have, then, for multiple sensor modalities, when the instruction (whether natural or formal language) can completely specify the task?

**Summary Of The Paper:**

The manuscript proposes 'latent goal architectures' for following formal instructions in OOD scenarios.

**Summary Of The Review:**

I have some issues with the novelty of the work presented, the accuracy of the claims made, and the somewhat lacking comparison with the related work and key ablation experiments. See main review, above.

---

> ### Author Response · Authors · 2021-11-18
> **Response to reviewer FvKn (1/2)**
>
> We thank the reviewer for their detailed feedback. Nonetheless, we would like to point to some possible misunderstandings that might have negatively impacted the reviewer's assessment.
>
> > Section 3.2: This approach, especially with the definition of the symbolic module, seems to bypass many of the challenging problems ... The SM provides a lot of privileged information, through its labelling function and comprehensive reward function.
>
> We agree with the reviewer that problems such as sparse rewards or noisy instructions are of great interest, but these are out of the scope of this work. Nevertheless, our paper focus on the ability of agents to generalise to zero-shot instructions in out-of-distribution environments, both very challenging and relevant problems on their own (Section 1).
>
> Please note that the SM does not have any privileged information, what it uses is the same information used in standard RL literature:
> * The reward function is equal to a standard reward function of a POMDP in RL, but instead of having the human engineer defining a fixed reward function for the environment, it is engineered to give rewards according to the human instruction.
> * The labelling function is just an "event detector", e.g., the same way that you need an event detector to give a penalisation to your car if it is over the speed limit, we need this labelling function to tell when the agent reached the key and provide a positive reward if the goal was to reach the key.
>
> We updated Section 3.2, where with additional clarifications to avoid confusions (see also Algorithms 1-3 in the Appendix for further detail).
>
> > Section 3.2: How is ambiguity handled — i.e., when there exists multiple ways that T can be entailed?
>
> When there are multiples paths to follow the neural module is given the different choices through disjunctions in the goals, once one of the goals was satisfied the other path is discarded (Algorithms 2 and 3 in the Appendix)
>
> > Section 3.3: The manuscript states "Consequently, it can be beneficial for agents to compute both human instructions and sensory inputs in a dedicated channel, generating a latent representation of the current goal. We call this type of deep learning model a latent-goal architecture." By itself, this is not novel: there are a plethora of works in learning robot skills, and they have dramatically less supervision
>
> We agree with the reviewer that there are works computing instructions and sensory inputs together (see Section 6, last paragraph) but they do it in a different way that our neural network architecture does, which is novel.
>
> Regarding the concern about other works having less supervision, we believe this is motivated by the misinterpretation we responded earlier about the labelling function and rewards. Nevertheless, we include works about robot skills (last paragraph Section 6), and highlight Karkus et al. (2020), whose framework generates goals given observations and human instructions. Nevertheless, note that they require further supervision that our approach since they require human-designed goals from which the neural network learns to select the most relevant at each step. Also, they do not study inductive biases that we demonstrate key in our work (e.g. the information bottleneck). Our network is end-to-end, and the latent goals are learnt and generated implicitly without any human intervention.
>
> > Section 4: Table 1: Train/Test results should be compared separately. Bolding schemes (to highlight the best-performing model) change, from table to table; this should be made consistent.
>
> Thank you for the pointer, you can see in the updated version (Section 5) that we substituted all the tables with bar charts, performed a 2-way ANOVA test of table 1 as suggested by Reviewer Ymus, and separated evaluations for train and test results. See Appendix D for even further discussion.
>
> > Section 4: The manuscript somehow makes reference to neither the robot skills literature nor the robot instruction-following literature. Additional experiments and comparisons are needed.
>
> Please note that our work is in the broad field of agents following human instructions. Nevertheless, as stated above, two references that we contrast our work with in Section 6 come from the robot instruction-following literature (Huang et al. (2020) and Karkus et al. (2020)). We are happy to include further relevant literature the reviewer might find.
>
> Additionally, note that our experiments are meant to motivate that works such as those in the robot literature, adopt a latent goal architecture in exchange of standard neural network configurations.
>
> We continue in a response below with the last points

---

> > ### Author Response · Authors · 2021-11-18
> > **Response to reviewer FvKn (2/2)**
> >
> > > Throughout: The manuscript relies on the assumption that "human instructions can be separated from the sensory input". This is a strong and unrealistic assumption, in practice.
> >
> > Note that the literature of autonomous agents following human instructions typically works with agents receiving instructions through a separated text input, (e.g. all the works that use MiniGrid as a benchmark, see also Sections 1 and 6).  We believe that it is not unrealistic to think that future multi-module agents can synthetize the human instructions and separate or mask it from the rest of the observation.
> >
> > Please also note that, to demonstrate that our architecture improves performance with multiple instruction formats, we evaluate the agents with both visual and text instructions. Nevertheless, to make a fair comparison between the two environments, we need to make the same assumption of being able to separate the instructions in both. We added a clarification note in Sec. 5.
> >
> >
> > We believe that our answers responded to all the original concerns from the reviewer. Nevertheless, we are happy to clarify any last-minute question/concern the reviewer might have.

---

> > > ### Author Response · Authors · 2021-11-22
> > > **Follow up**
> > >
> > > Please let us know if the reviewer is satisfied with our response and updated draft, note that the response period for authors ends today. We are happy to address any further questions.

---

> > ### Comment · Reviewer_FvKn · 2021-12-01
> > **Thanks**
> >
> > I appreciate the authors' responses. They did clarify some of my points, however, I remain unconvinced on others:
> >
> > Re: Scope + Motivation. I re-read Section 1, following the updates, and still had the same concerns on motivation. The manuscript and author response seek to set aside the challenges faced in common robot instruction-following settings (sparse rewards, no explicit progress-monitoring, noisy instructions, noisy environment, no task-specific reward-shaping, goal/initialisation mismatch) as being out of scope. Remembering that the reasons why the community shifted to natural language (NL) instruction-following are that (i) NL is closer to the semantic level of granularity that agents would expect in human-robot interaction contexts and that (ii) NL is simply easier to crowdsource, compared to more formal task specification, the manuscript inherits the responsibility of motivating why formal specification is still reasonable. While a reader may appreciate the detailed formulation provided by the manuscript, some may be left wondering how the work described can be of practical benefit to the field.
> >
> > Re: reward function: the reward function described by the manuscript provides much more (dense) supervision than in popular POMDP instruction-following settings. See references below.
> >
> > Re: labelling function: in instruction-following tasks, labeling functions (or "event detectors", sure) that tell the ego-agent when it has completed a task are commonly referred to as oracles, and the information they provide is most certainly privileged. How can an agent learn when/where to stop executing, if it enjoys the convenience of such an "event detector"? How can an agent handle complex tasks in unseen environments, if it has not learned how to estimate task progress/completion on its own?
> >
> > My earlier point was that task-execution in this work, despite what the manuscript refers to as "unseen instructions in OOD environments", benefits greatly from the formal task specification (avoids the aforementioned challenges in instruction-following) and increased supervision (dense rewards, labeling functions), compared to popular settings. Despite carefully considering the manuscript updates and author responses, I am unable to increase my score.
> >
> > Re: robot skills literature:
> >
> > https://ieeexplore.ieee.org/abstract/document/9196582
> >
> > https://ieeexplore.ieee.org/stamp/stamp.jsp?arnumber=8462891
> >
> > https://openreview.net/pdf?id=rk07ZXZRb
> >
> > https://arxiv.org/pdf/1810.02422.pdf
> >
> > Re: instruction-following literature:
> >
> > https://openaccess.thecvf.com/content_cvpr_2018/papers/Anderson_Vision-and-Language_Navigation_Interpreting_CVPR_2018_paper.pdf
> >
> > https://openaccess.thecvf.com/content_CVPR_2019/papers/Wang_Reinforced_Cross-Modal_Matching_and_Self-Supervised_Imitation_Learning_for_Vision-Language_Navigation_CVPR_2019_paper.pdf
> >
> > https://openreview.net/pdf?id=r1GAsjC5Fm
> >
> > https://arxiv.org/abs/2106.13948 (a survey)

---

> > > ### Author Response · Authors · 2021-12-02
> > > **Thank you and response**
> > >
> > > > Re: Scope + Motivation.
> > >
> > > We thank the reviewer for the additional time spent with our work. While the reviewer mentions very interesting and challenging problems from the robotics and locomotion literature, e.g. learning from human demonstrations, please note that our work is focused on the problem of agents learning compositionally from human instructions in an RL framework, which is also a very challenging problem with a well-established and recognised community [1-8]. In our line of work, the assumptions we have made are non-controversial. Nonetheless, note that our architecture and the inductive biases we propose only assume that instructions can be separated from the input. We note that the references you provided on following human instructions also work on that setting. Thus, our contributions can also be applied to this literature. We will incorporate the most relevant references in the final version of our paper.
> > >
> > > > NL and formal specifications
> > >
> > > While expressing instructions in temporal logic can be beneficial (e.g. its unambiguous semantics and compositional syntax allows for easier procedural generation and evaluation of instructions), please note that none of the novel features of our neural network architecture prevents it from being used with NL instructions instead of temporal logic. Also, currently there is thriving research in RL and DRL agents working with temporal logic instructions and related automata, including reward machines [6-8] (Section 1).
> > >
> > > > Re: reward and labelling function
> > >
> > > Please note that our reward and labelling function follows the same assumptions of previous literature from our community [1-8], that is, providing a reward or penalisation when agents satisfy or violate a human instruction, respectively.
> > >
> > > > My earlier point was that task-execution in this work, despite what the manuscript refers to as benefits greatly from the formal task specification, and increased supervision (dense rewards, labelling functions), compared to popular settings.
> > >
> > > Please note that our architecture and the 6 baselines (MLP, RelNet, MHA, PrediNet, BRIMs and ResBRIMs) are all evaluated in the same setting. The benefits mentioned by the reviewer are given to all SOTA architectures we are comparing our proposed configuration with. Also, the settings we use (the procedural Minecraft-inspired maps and MiniGrid), are very popular in the field of DRL agents learning compositionally from human instructions (we refer to Section 5 for detail, but for ease of reading https://github.com/maximecb/gym-minigrid keeps a list of some of the latest works using this benchmark).
> > >
> > > [1] Junhyuk Oh, Satinder Singh, Honglak Lee, and Pushmeet Kohli. Zero-shot task generalization with multi-task deep reinforcement learning. In Proceedings of the 34th International Conference on Machine Learning-Volume 70, pp. 2661–2670. JMLR. org, 2017.
> > >
> > > [2] Dzmitry Bahdanau, Shikhar Murty, Michael Noukhovitch, Thien Huu Nguyen, Harm de Vries, and Aaron C. Courville. Systematic generalization: What is required and can it be learned? In 7th International Conference on Learning Representations, ICLR 2019, New Orleans, LA, USA, May 6-9, 2019. OpenReview.net, 2019.
> > >
> > > [3] Chevalier-Boisvert, M., Bahdanau, D., Lahlou, S., Willems, L., Saharia, C., Nguyen, T. H., & Bengio, Y. (2018). Babyai: A platform to study the sample efficiency of grounded language learning. In 7th International Conference on Learning Representations, ICLR 2019, New Orleans, LA, USA, May 6-9, 2019. OpenReview.net, 2019.
> > >
> > > [4] Brenden M Lake. Compositional generalization through meta sequence-to-sequence learning. In Advances in Neural Information Processing Systems, pp. 9788–9798, 2019.
> > >
> > > [5] Felix Hill, Andrew Lampinen, Rosalia Schneider, Stephen Clark, Matthew Botvinick, James L McClelland, and Adam Santoro. Environmental drivers of systematicity and generalization in a situated agent. In International Conference on Learning Representations, 2020.
> > >
> > > [6] Rodrigo Toro Icarte, Ethan Waldie, Toryn Klassen, Rick Valenzano, Margarita Castro, and Sheila McIlraith. Learning reward machines for partially observable reinforcement learning. In Advances in Neural Information Processing Systems, pp. 15497–15508, 2019.
> > >
> > > [7] Pashootan Vaezipoor, Andrew Li, Rodrigo Toro Icarte, and Sheila McIlraith. Ltl2action: Generalizing ltl instructions for multi-task rl. International Conference of Machine Learning, 2021.
> > >
> > > [8] Yen-Ling Kuo, Boris Katz, and Andrei Barbu. Encoding formulas as deep networks: Reinforcement learning for zero-shot execution of LTL formulas. IEEE/RSJ International Conference on Intelligent Robots and Systems, IROS 2020, Las Vegas, NV, USA, October 24, 2020 - January 24, 2021, pp.5604--5610, IEEE, 2020.

---

### Official Review · Reviewer_bGYm · 2021-11-05

**Correctness:** 4
**Technical Novelty And Significance:** 3
**Empirical Novelty And Significance:** 3
**Recommendation:** 8
**Confidence:** 3

**Main Review:**

In my opinion, this paper is well laid out and tackles the important problem of creating agents that can perform high level temporal tasks. While a fairly incremental contribution, the thorough empirical analysis indicates that performing out of distribution temporal tasks is possible.

One thing I was surprised was not explicitly done was to use the underlying automata structure of proposed reward function to further accelerate the RL loop. In particular, I recent work on reward machines (which can encode regular languages) uses a trick similar to hindsight experience replay to provide counterfactual examples that and mitigate reward sparsity issues. Looking at the training plots in the appendix, I wonder if the generalization may have occurred if many fewer steps. I am also curious if such a trick could be used to remove the small (and seemingly adhoc) penalization incurred at each time step (which is in theory redundant with the discount rate, but a source of possible reward bugs).

**Summary Of The Paper:**

This work focuses on training (via deep RL) task conditioned agents for a given class of workspaces. Tasks are formalized as a variant of finite temporal logic (and thus encode a regular language). Tasks are either provided as text or via visual diagrams. A key goal of this work is to perform well on tasks that lie outside of the training distribution. The paper illustrates on two common grid like domains with the results of a number of various architectures and ablations reported.

The approach taken is an adaptation of deep reinforcement learning, with the primary contribution being a slight, but seemingly important, modification of an existing architecture to create a separate embedding for (i) a task agnostic state encoding and (ii) create a task specific state encoding. Furthermore, in order to encourage the network to generalize between tasks, an information bottleneck is placed on the task specific encoding.

**Summary Of The Review:**

Overall, while somewhat incremental, I think this paper adds a valuable contribution, is very thorough, and pretty coherent. The focus on simply modifying the inductive biases in the architecture (rather than dramatically changing training) is also a good venue fit.

---

> ### Author Response · Authors · 2021-11-18
> **Response to Reviewer bGYm**
>
> We would like to thank the reviewer for their positive feedback. In particular for highlighting the thoroughness and pretty coherence of our work. We respond to their questions below.
>
> > One thing I was surprised was not explicitly done was to use the underlying automata structure of proposed reward function to further accelerate the RL loop. In particular, recent work on reward machines (which can encode regular languages) uses a trick similar to hindsight experience replay to provide counterfactual examples that and mitigate reward sparsity issues.
>
> We agree with the reviewer that using reward machines within the symbolic module would be an interesting line to follow. However, no work with reward machines have used that framework to solve zero-shot atomic tasks, which is our main concern here. For this reason, we used the framework from Leon et al. (2020), which is --to the best of our knowledge-- the only work that explicitly studies this kind of generalisation with instructions in temporal logic. Our symbolic module differs from the one in Leon et al. (2020) only in that it is adapted to handle the more expressive instructions of SATTL instead of the TTL instructions from that previous work (Section 3.2).  We believe this is beneficial since it allows to keep the main focus and discussion on the neural part, the novel architectures and the numerous tests and ablations, which are our main contributions.
>
> >  Looking at the training plots in the appendix, I wonder if the generalization may have occurred if many fewer steps
>
> The number of training steps needed by our agents is similar to previous works tackling compositional learning and systematic generalisation (e.g., Hill et al. 2020, Leon et al. 2020). Nevertheless, we agree with the reviewer that this could be improved for instance by exploring different algorithms from A2C. However, this is out of the scope of this work which focuses on the impact of  different neural networks in an RL framework.
>
> > I am also curious if such a trick could be used to remove the small (and seemingly adhoc) penalization incurred at each time step (which is in theory redundant with the discount rate, but a source of possible reward bugs).
>
> Note that the small penalisation helps to avoid agents that may converge to local minima policies, e.g., a policy "move towards a wall" that tries to avoid the larger penalisations that may occur because of navigating through the environment. Additionally, since maps are procedurally generated, it may happen that the map generated requires the agent to violate the constrain to reach the goal, the small penalisation encourages the agent to do that (but only if there is not an alternative path).

---

### Official Review · Reviewer_Ymus · 2021-11-05

**Correctness:** 4
**Technical Novelty And Significance:** 3
**Empirical Novelty And Significance:** 4
**Recommendation:** 8
**Confidence:** 4

**Main Review:**

**Major Comments**

*Sound policy network design choices:* The design choices in the proposed latent goal representations appear to be sound. It appears that joint training of the goal specific information with the task embedding is beneficial to generalization. And this design choice appears to be benefiting the performance irrespective of the network architecture of the central module. However, this is hard to judge considering the presentation of the results.

*Training and testing on multiple formulas:* I like the fact that the authors have sampled multiple instructions to test on and not on a few handpicked formulas. The authors should provide the definition of the sampling algorithm for the instructions in the appendix.

*Presentation of the results:* My biggest issue with the paper is how the results are presented and analyzed. I urge the authors to use bar charts instead of raw numbers to present the results as the comparisons are much easier for the readers. Further I believe that the standard errors of many of the conditions overlap with each other. The authors should perform statistical analysis appropriately before claiming superiority of any of the conditions over the others. With so many experiments, and data points, sound statistical analysis is important. a 2-way ANOVA (independent variables being network architecture, and latent goal representation or the classical architecture) with repeated measures over the minigrid and minecraft domain seem to be the appropriate statistical approach. Without this, the claims of benefits of one over the other are tenuous. While the authors can include this in the final version, I believe the authors should be able to run this before the responses are due.

**Minor comments:**
Please consider the following suggestions for clarity:
1. I would suggest the authors to bulletize the baseline architectures, instead of writing a long paragraph. That would make it easier to parse the important features.
2. Separate discussion of the results from the ablation studies, right now you only get to the ablations at the end of that subsection
3. I would have liked to see some contrastive description of SATTL with respect to TTL. How does SATTL improve the expressivity of TTL? What types of task cannot be expressed in TTL but can be in SATTL?




**Summary Of The Paper:**

This paper lies in the space of using non-Markov reward functions to model temporal task, and in addition to that attempts to generalize instruction following beyond single instruction provided in a formal language. The authors propose a dual network solution to embed the environment representation and separately a goal specific reasoner grounded within the environment. For planning with temporal formulas, the authors extend prior work on Task temporal logic. Although the algorithm for this remains largely unchanged from prior work (please clarify if this is wrong).

The results demonstrate that the latent goal representation helps in generalizing at both the object level and at the task level. However the presentation of the results can be significantly improved (please refer to the main review).

**Summary Of The Review:**

I believe that as written the paper is not ready for publication. I believe that statistical analysis of the results is necessary, and can be reasonably completed in the response period. I would be willing to upgrade my scores (correctness and significance) based on that. I also believe that the edits for clarity would significantly strengthen the paper.

Update: The authors incorporated most of the comments, and the statistical analysis bears out the major claims.

---

> ### Author Response · Authors · 2021-11-18
> **Response to Reviewer Ymus (1/2)**
>
> We thank the reviewer for the feedback. We really appreciate that the reviewer critically highlights the sound design choices and our empirical testing. We agree with the reviewer that their suggested changes improve the clarity of the paper and the confidence of the empirical results. Thus, **the updated version contains all the reviewer's suggestions**. Also that the 2-way ANOVA statistical analysis confirms the positive results about the impact of the latent-goal architecture in both Minecraft and MiniGrid, specially with unseen instructions. Below we provide a detailed response to all the points raised by the reviewer:
>
> > The authors should provide the definition of the sampling algorithm for the instructions in the appendix.
>
> The updated version includes now Algorithm 4 in Appendix B and explains how instructions are generated. Intuitively, the algorithm randomly selects a type of task (task types are described in Appendix B in MiniGrid paragraph), and then selects an "eligible" set objects from the corresponding train or test set of objects for that type of task (to generate train or test tasks, correspondingly). Note that in Minecraft the set of test objects are the same for all task types since in that environment all test instructions refer to zero-shot objects. Then, the algorithm randomly selects the number of goals and safety conditions that the task will have in this episode and simply selects different objects from the eligible set for every goal and safety condition.
>
> Note that to further facilitate reproducibility we will include a link to the code in the final version of the paper if it is accepted.
>
> > I urge the authors to use bar charts instead of raw numbers to present the results as the comparisons are much easier for the readers
>
> Thank you, we updated that. We agree that this makes easier for the reader to see the impact of the proposed approach. Still, we kept the original tables in the Appendix, in case some reader prefers that format to see the specific values.
>
> > The authors should perform a 2-way ANOVA (independent variables being network architecture, and latent goal representation or the classical architecture)
>
> As we indicate now in Section 5, in the paragraph of "Results with multi-layered architectures" we added the results of a 2-way ANOVA test in Appendix D.2. We include the ANOVA test results in a separated response below
>
> > I would suggest the authors to bulletize the baseline architectures, instead of writing a long paragraph. That would make it easier to parse the important features.
> > Separate discussion of the results from the ablation studies, right now you only get to the ablations at the end of that subsection
>
> Thank you, fixed.
>
> > I would have liked to see some contrastive description of SATTL with respect to TTL. How does SATTL improve the expressivity of TTL? What types of task cannot be expressed in TTL but can be in SATTL?
>
> This is a good point, we included a constrative description of the two languages in the first paragraph of Section 3.1. The main difference is the inclusion in SATTL of the "Until" operator. Intuitively TTL allows only for reachability goals, e.g. "eventually reach a sword", while SATTL can express additional constraints, e.g., "walk on soil or stone until you reached a sword".

---

> > ### Author Response · Authors · 2021-11-18
> > **Response to Reviewer Ymus (2/2)**
> >
> > Here we include the results of the 2-way ANOVA, first the different tables then the analysis in Appendix D.2
> >
> > Note that  "architecture configuration" refers to whether using latent goals or not and "neural network" to the type of network used in the central module. Alpha was set to 0.05.
> >
> > |  Minecraft-Train 	|  F	|   P-value	|   F-critic	|
> > |----	|:---:	|:---:	|:---:	|
> > |  Architecture configuration	|   43.56 	|   2.74exp−10	|  3.88 	|
> > |  Neural Network 	|    237.31	|   2.13exp−70	|  2.64 	|
> > |   Interactions	|  31.27  	|   5.16exp−17	|   2.64	|
> >
> > |  Minecraft-Test 	|  F	|   P-value	|   F-critic	|
> > |----	|:---:	|:---:	|:---:	|
> > |  Architecture configuration	|   23.63	|   2.15exp−6	|  3.88 	|
> > |  Neural Network 	|    166.75	|   1.26exp−57	|  2.64 	|
> > |   Interactions	|  16.26  	|   1.26exp−09	|   2.64	|
> >
> > |  MiniGrid-Train 	|  F	|   P-value	|   F-critic	|
> > |----	|:---:	|:---:	|:---:	|
> > |  Architecture configuration	|   1.37 	|   0.24	|  3.93 	|
> > |  Neural Network 	|    39.57	|   1.63exp−17	|  2.69 	|
> > |   Interactions	|  1.86  	|   0.14	|   2.69	|
> >
> > |  MiniGrid-Test 	|  F	|   P-value	|   F-critic	|
> > |----	|:---:	|:---:	|:---:	|
> > |  Architecture configuration	|   8.57	|   0.004	|  3.93 	|
> > |  Neural Network 	|    5.51	|   0.001	|  2.69 	|
> > |   Interactions	|  0.53 	|   0.6	|   2.69	|
> >
> > To confirm that the standard error of the different conditions do not overlap each other, we perform a two-way analysis of variance (two-way ANOVA) with the independent variables being the neural networks used in the central modules and the architecture configuration, i.e., latent goal (ours) or standard configuration. The 2-way ANOVA is done across the maps of different size of Minecraft and MiniGrid respectively. Note that to evaluate the different hypotheses, we need to check whether the F value is greater than its corresponding F-critic or not, and to confirm that the P-value is smaller than Alpha.
> >
> > Tables "Mineacraft-Train and Minecraft-Test" show the results of the ANOVA analysis with the results from Minecraft. We see that it confirms that both the architecture configuration and the neural network have a significant impact of the performance of the agent. Noticeably, the ANOVA results show that there is not a significant interaction between the two variables, meaning that the neural network is not relevant in the general improvement in performance that latent goals grant in this setting. Still, note that this is analysis of the statistical impact averaged across all the networks. As studied in Sec. 5.1 the latent goals do have a different impact with the PrediNet than with other networks.
> >
> > Tables "MiniGrid-Train and MiniGrid-Test" show the results of the ANOVA test with MiniGrid. Here we see that neural networks have a significant impact in the training performance of the agent but that is not the case with the architecture, i.e., using latent goals does not have a significant impact in the global training performance. Nevertheless, we see that this changes with unseen instructions (test) where the latent goals have a stronger impact in performance than the neural network, being both statistically relevant. Last, from the interactions' results we see that both in training and test the interaction of the two variables have a significant impact in the final performance, i.e., the effect in performance that latent goals have is strongly dependant on the neural network and vice versa.
> >
> > Regarding the differences in how network and architecture impact in Minecraft and MiniGrid, we believe that these are motivated by the additional difficulty of MiniGrid observation and action settings. Specifically, MiniGrid only allows agents to move forward or turn around, and agents can only observe what is in front of them. This requires agents to rely more on their memory than they do in Minecraft, where they can observe the objects around them in any direction and have actions to move in four different directions. This point also motivated the need of using an schedule in the introduction of larger training maps in MiniGrid, as detailed in Appendix D.3

---

> > > ### Author Response · Authors · 2021-11-22
> > > **Follow up**
> > >
> > > Please let us know if the reviewer is satisfied with our response and updated draft, note that the response period for authors ends today. We are happy to address any further questions.

---

### Author Response · Authors · 2021-11-18
**Updated Abstract**

Following Reviewer 2S8A suggestion, we updated the abstract but we cannot modify it in the text above. We copy below the updated version:


*We address building agents whose goal is to execute out-of distribution (OOD) multi-task instructions expressed in temporal logic (TL) by using deep reinforcement learning (DRL). Recent works provided evidence that the agent's neural architecture is a key feature when DRL agents are learning to solve OOD tasks in TL. Yet, the studies on this topic are still in their infancy. In this work, we propose a new deep learning configuration with inductive biases that lead agents to generate latent representations of their current goal, yielding a stronger generalisation performance. We use these latent-goal networks within a neuro-symbolic framework that executes multi-task formally-defined instructions and contrast the performance of the proposed networks against employing different state-of-the-art (SOTA) architectures when generalising to unseen instructions in OOD environments.*

---

### Decision · Program_Chairs · 2022-01-20

**Decision:**

Accept (Poster)

**Comment:**

The paper considers the problem of learning to carry out novel, multi-task instructions specified via temporal logic using deep reinforcement learning. A specific focus of the paper is improving generalization to test-time instructions that differ from those encountered during training. To facilitate this generalization, the proposed architecture encodes a latent specification of the goal according to the given instruction and environment state that is then combined with a task-agnostic environment embedding. Experiments on grid-like domains demonstrate that the proposed framework outperforms recent deep RL approaches to satisfying temporal logic-based instructions.

The instruction-following problem has long been of interest in the robotics, ML, and broader AI communities dating back several decades. The problem has received renewed attention in the last few years, largely as a target for neural network-based multi-view and RL learning architectures. The primary contribution of this paper is the proposed extension of existing deep RL approaches to reason over a learned, latent goal specification as a means of improving generalization to novel test-time utterances. The approach is sound and several reviewers agree that the ablation studies together with comparisons to contemporary deep RL architectures support the advantage of these inductive biases. The reviewers raised initial concerns regarding the statistical significance of the results and the clarity of the presentation. The authors provided detailed feedback to the reviewers and updated the paper to address many of these concerns, largely satisfying two of the reviewers.

However, concerns remain that the paper doesn't adequately position this work in the context of the decades worth of research in instruction-following. Early work in this area focused on interpreting highly structured instructions (e.g., formal logic-based), first using rule-based methods, and then parsers trained via supervised learning. Over the past decade, however, the field has largely moved towards learning to follow instructions conveyed in "natural" language, which brings with it a significant number of challenges, including the assumption that test-time instructions will inherently be out-of-distribution. That is not to say that the contributions of the paper aren't interesting---they are, but in the relatively narrow scope of deep RL-based approaches to following structured, temporal logic instructions.